# ReDistill: Residual Encoded Distillation for Peak Memory Reduction of CNNs

**Fang Chen**[1]                                          *fchen20@ucmerced.edu*

**Gourav Datta**[2]                                       *gourav.datta@case.edu*

**Mujahid Al Rafi**[1]                                      *mrafi@ucmerced.edu*

**Hyeran Jeon**[1]                                         *hjeon7@ucmerced.edu*

**Meng Tang**[1]                                          *mtang4@ucmerced.edu*

[1] *University of California Merced*
[2] *Case Western Reserve University*

**Reviewed on OpenReview:** *https://openreview.net/forum?id=akumIxQjNN*

## Abstract

The expansion of neural network sizes and the enhanced resolution of modern image sensors result in heightened memory and power demands to process modern computer vision models. In order to deploy these models in extremely resource-constrained edge devices, it is crucial to reduce their peak memory, which is the maximum memory consumed during the execution of a model. A naive approach to reducing peak memory is aggressive down-sampling of feature maps via pooling with large stride, which often results in unacceptable degradation in network performance. To mitigate this problem, we propose residual encoded distillation (ReDistill) for peak memory reduction in a teacher-student framework, in which a student network with less memory is derived from the teacher network using aggressive pooling. We apply our distillation method to multiple problems in computer vision, including image classification and diffusion-based image generation. For image classification, our method yields **4x-5x** theoretical peak memory reduction with less degradation in accuracy for most CNN-based architectures. For diffusion-based image generation, our proposed distillation method yields a denoising network with **4x** lower theoretical peak memory while maintaining decent diversity and fidelity for image generation. Experiments demonstrate our method's superior performance compared to other feature-based and response-based distillation methods when applied to the same student network. The code is available at https://github.com/mengtang-lab/ReDistill.

## 1 Introduction

Convolutional neural networks (CNN) have demonstrated impressive capabilities across diverse computer vision tasks such as image recognition (Simonyan & Zisserman (2014)), object detection (Redmon & Farhadi (2018)), semantic segmentation (Long et al. (2015)), and image generation (Creswell et al. (2018)). However, the ever-growing network size and image resolution of modern imaging sensors pose significant challenges in deploying neural networks on standard edge devices with limited memory footprint. For example, a standard STM32H5 MCU provides only 640 KB of SRAM and 2 MB of Flash storage. These constraints make it impractical to execute off-the-shelf deep learning models: ResNet-50 surpasses the storage limit by 44×, while MobileNetV2 exceeds the peak memory limit by 8×. Even the int8 quantized version of MobileNetV2 surpasses the memory limit by 2×, underscoring a substantial disparity between desired and

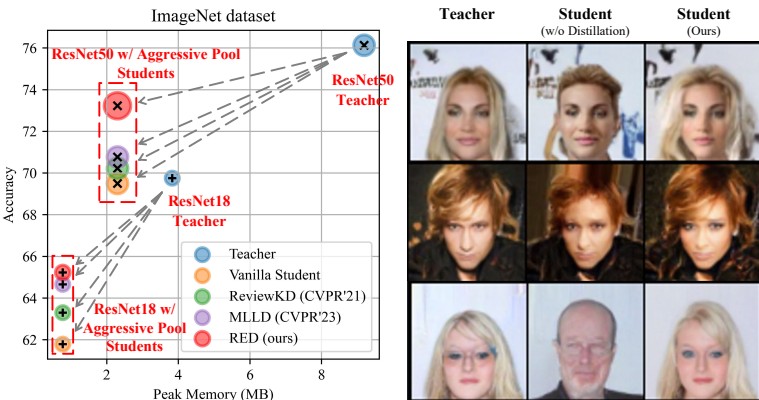

Figure 1: (a) *Left*: For ImageNet classification, our distillation method significantly reduces the theoretical peak memory of ResNet-based models while achieving accuracy better than existing distillation methods. (b) *Right*: For diffusion-based image generation, our distilled network with 4× lower theoretical peak memory generates images indistinguishable from the generated images of a teacher network.

available hardware capacity. Hence, it is very important to reduce the *peak memory* during inference for edge deployment. Note that our primary focus in this work is reducing peak memory usage, as there are existing solutions for addressing other metrics, such as parameter count and the number of operations when deploying CV models at the extreme edge. Similar to (Lin et al. (2021); Chowdhery et al. (2019)), we estimate the theoretical peak memory by summing the size of the input & output allocation for each operation (e.g., convolution, non-linear activation, pooling). Through empirical measurements, we determined that peak memory usage is predominantly influenced by the initial layers of convolutional neural networks (CNNs) that are characterized by large feature maps. For U-shaped CNN architectures, however, the last few layers also significantly contribute to peak memory consumption.

A naive approach to reducing peak memory is aggressive downsampling via pooling with large kernel size and large stride, which often leads to unacceptable degradation of network performance due to loss of information in small feature maps. Given a teacher network with large peak memory, we propose residual encoded distillation (ReDistill) to train a student network with significantly lower peak memory. This student network can be considered a variant of the teacher network with aggressive pooling. We demonstrate the effectiveness of our methods for multiple problems including image classification and diffusion-based image synthesis. For image classification with ResNet-based models shown in Fig. 1 (a), our method reduces the theoretical peak memory by $4-5\times$ with a lower accuracy drop compared to existing distillation methods. For diffusion-based image generation shown in Fig. 1 (b), our distilled network generates images similar to original networks, yet the theoretical peak memory is reduced by $4\times$ on average.

Our ReDistill method outperforms existing response-based or feature-based distillation methods regarding the accuracy-memory trade-off. Our method differs from these existing counterparts in four regards. Firstly, our distillation method is tailored for peak memory reduction. In contrast, existing distillation techniques focus on transferring knowledge from a high-capability teacher network with a large number of parameters to a student network with fewer parameters. Our student networks apply a large kernel size and stride in the initial pooling layers with the same number of parameters as the teacher networks while consuming significantly lower peak memory. Secondly, the student network, utilizing aggressive pooling, has fewer pooling layers and consequently fewer stages than the teacher network, resulting in mismatched features at different stages between the two. We add novel non-linear mapping modules termed residual encoded distillation (RED) blocks between the teacher and student network during both training and inference. Thirdly, our proposed RED block is lightweight and effective with additive residual learning and multiplicative gating mechanism. We optimize the trade-off between peak memory and accuracy, while it slightly increases the model size due to extra parameters. Lastly, we align teacher and student network features asynchronously at pooling layers with matching feature sizes, while previous approaches align features at different stages of the networks.

Our key contributions are summarized below.

- We propose ReDistill, a distillation framework tailored for reducing the peak memory of convolutional neural networks. Our method allows aggressive downsampling of feature maps via pooling layers with a large stride for a student network while incurring a less accuracy drop. To the best of our knowledge, ReDistill is the first distillation method focused on peak memory reduction for efficient deep learning.

- The core of our ReDistill framework is a residual encoded distillation (RED) block to align features between high-peak-memory teacher networks and low-peak-memory student networks. Our RED block is based on a multiplicative gating mechanism and additive residual learning and is shown to be simple and effective for peak memory reduction with minimum computational overhead.

- For image classification tasks, our distillation method outperforms state-of-the-art response-based or feature-based distillation methods when applied to the same student network assigned with a large pooling stride, as shown in extensive experiments with multiple datasets. Our method yields $4\times \sim 5\times$ reduction in theoretical peak memory with a slight decrease in classification accuracies for CNN based models.

- We also show the versatility of our distillation method for denoising diffusion probabilistic models for image generation. For a U-Net based denoising network, our method reduces the theoretical peak memory by $4\times$ by downsampling the feature maps of the first few encoder layers and last few decoder layers while maintaining the fidelity and diversity of synthesized images.

## 2 Related Work

**Memory-constrained deep learning** Limited memory capacity in GPU cards and edge platforms has been a critical hurdle in CNN training and inference. Multiple GPUs can be utilized through model and data parallelism (Langer et al. (2020)) to mitigate the memory bottleneck. Other solutions include optimization methods such as network quantization (Hubara et al. (2016)), compression (Han et al. (2016b)), and pruning (Molchanov et al. (2017); Lu et al. (2024)), which focus on maintaining essential bits of weights or parameters while minimizing accuracy loss. To produce correct outputs with compressed data, these solutions are typically designed with specialized accelerators to accommodate meta-data processing (Han et al. (2016a)). There are also CNNs specifically designed for resource-constrained applications, such as variants of MobileNet (Howard et al. (2019)) and SqueezeNet (Iandola et al. (2017)). These approaches to memory-constrained deep learning are orthogonal to our ReDistill framework, which focuses on peak memory reduction during inference. Nevertheless, recent work has explored neural architecture search (NAS) to create networks with minimized peak memory (Lin et al. (2020)). Reference (Lin et al. (2021)) takes this a step further by leveraging NAS to introduce patch-based inference and network redistribution (Lin et al. (2021)), consequently shifting the receptive field to later stages. While NAS significantly exacerbates the training complexity, patch-based inference necessitates compiler libraries that may not be compatible with standard GPUs and incurs additional computation and latency overhead. Another recent work (Chen et al. (2023)) proposed self-attention-based pooling to aggressively compress the activation maps in the first few layers to reduce the peak memory at the cost of increased compute complexity.

**Knowledge distillation for image classification** can be roughly categorized into two groups: response-based KD and feature-based KD. Response-based KD methods derive the distillation loss by leveraging the logit outputs from the fully connected layers of the student model and the teacher model. For example, KD (Hinton et al. (2015)) distills knowledge by matching the prediction probability distributions of the student architecture and the teacher architecture. DKD (Zhao et al. (2022)) decouples the classical KD loss into two parts, target class knowledge distillation (TCKD) and non-target class knowledge distillation (NCKD) enhancing training efficiency and flexibility. MLLD (Jin et al. (2023)) performs logit distillation through a multi-level alignment based on instance prediction, input correlation, and category correlation, delivering state-of-the-art performance. In contrast, feature-based KD methods (Adriana et al. (2015); Zagoruyko & Komodakis (2016); Passalis & Tefas (2018); Lee et al. (2018); Tung & Mori (2019); Ahn et al. (2019)) reduce the disparity between features in the teacher and student models, compelling the student model to replicate the teacher model at the feature level. RKD (Park et al. (2019)) employs a

relation potential function to convey information from the teacher's features to the student's features. ReviewKD (Chen et al. (2021)) aggregates knowledge of the teacher from different stages into one stage of the student, the so-called 'knowledge review', which achieved impressive performance. KCD (Li et al. (2022)) iteratively condenses a compact knowledge set from the teacher to guide the student learning by the Expectation-Maximization (EM) algorithm, which would empower and be easily applied to other knowledge distillation algorithms. Existing methods focus on the distillation from a high-capacity teacher model with a large amount of parameters to an efficient student model with limited parameters. In this work, the student model, employing a large kernel size and stride in the initial pooling layer possesses the same number of parameters as the teacher architecture but incurs significantly lower peak memory.

**Knowledge distillation for diffusion models** has gained popularity. For example, One Step Diffusion (Yin et al. (2023)) defines two score functions, one of the target distribution and the other of the synthetic distribution produced by a one-step generator. By minimizing the KL divergence between these two score functions, the one-step generator is enforced to match the diffusion model at the distribution level and achieves impressive performance. Adversarial Diffusion (Sauer et al. (2023)) utilizes both score distillation loss and adversarial loss. The score distillation loss occurs between the teacher diffusion sampler with a large number of $T$ steps and the student diffusion sampler with one or two steps. Meanwhile, the adversarial loss originates from a discriminator trained to differentiate between generated samples and real images. Auto Diffusion (Li et al. (2023)) searches for the optimal time steps and compressed models in a unified framework to achieve effective image generation for diffusion models. In summary, existing KD methods for diffusion models mainly focus on time-step reduction and model compression. We offer a unique and orthogonal approach by minimizing peak memory that can be easily integrated with existing methods.

## 3 Proposed Method

### 3.1 Preliminaries

**Knowledge Distillation** We are given a dataset $\mathcal{X}$, a high-capacity teacher architecture $\mathcal{T}$ and a to-learned student architecture $\mathcal{S}$. For an input image $x$ sampled from $\mathcal{X}$, $\pi_{\mathcal{T}}(x)$ and $\pi_{\mathcal{S}}(x)$ denote the outputs or intermediate features of the teacher and student, respectively. The knowledge distillation task aims to optimize the student's parameters $\hat{w}$:

$$\hat{w} = \arg\min_w \sum_{x \in \mathcal{X}} \mathcal{L}(\pi_{\mathcal{S}}(x; w), \pi_{\mathcal{T}}(x)), \tag{1}$$

where $w$ denotes the trainable weights of $\pi_{\mathcal{S}}$ and $\mathcal{L}$ denotes the loss function defined by different knowledge distillation methods. For instance, Hinton et al. (2015) defines $\pi_{\mathcal{S}}$ and $\pi_{\mathcal{T}}$ as the logit outputs (without applying the softmax function) of the student and the teacher, while $\mathcal{L}$ as the Kullback-Leibler divergence between $\pi_{\mathcal{S}}$ and $\pi_{\mathcal{T}}$ after applying the softmax function with temperature $t_p$:

$$\mathcal{L}_{KL} = \sum_{x \in \mathcal{X}} KL(softmax(\frac{\pi_{\mathcal{S}}(x)}{t_p}), softmax(\frac{\pi_{\mathcal{T}}(x))}{t_p}). \tag{2}$$

Some other methods (Zagoruyko & Komodakis (2016); Park et al. (2019); Chen et al. (2021)) define different $\pi_{\mathcal{S}}$ and $\pi_{\mathcal{T}}$, such as the intermediate activation maps from various stages of the student and teacher, or different $\mathcal{L}$, like the $p$-norm, to achieve various distillation methods.

KD methods can be categorized as response-based methods with $\pi_{\mathcal{S}}, \pi_{\mathcal{T}}$ defined as the logit outputs, or feature-based methods with $\pi_{\mathcal{S}}, \pi_{\mathcal{T}}$ defined as the intermediate activation maps. Response-based KD methods generally keep the same peak memory with the student model, since they don't alter the student architecture. For feature-based KD methods, some might increase the student's peak memory due to extra trainable modules. However, none of these KD methods can lead to peak memory lower than the student, which is a lower bound. Our method reaches such a lower bound and achieves the highest accuracy compared to existing KD methods, as shown in our comprehensive experiments.

**Denoising Diffusion Probabilistic Models** Diffusion models (Ho et al. (2020)) are latent variable models of the form $p_\theta(x_0) := \int p_\theta(x_{0:T})dx_{1:T}$, where $x_1, ..., x_T$ are latents of the same dimensionality as the data

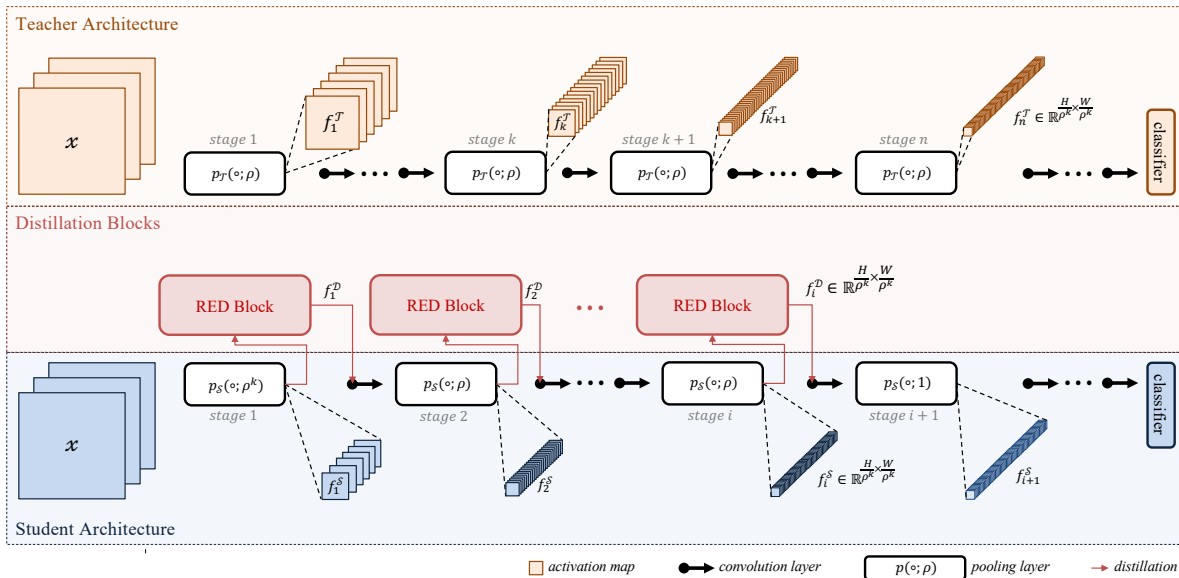

Figure 2: Our proposed residual encoded distillation framework (ReDistill). RED blocks are incorporated into the student model following the pooling layers to minimize the discrepancy between the down-sampled features of the student and teacher models.

$x_0 \sim q(x_0)$. The joint distribution $p_\theta(x_{0:T})$ is defined as a Markov chain with learned Gaussian transition starting at $p(x_T) = \mathcal{N}(x_T; \mathbf{0}, \mathbf{I})$, where $T$ is the maximum time step. In the training process, we are given a noisy input $x_t$, which is derived from the data $x_0$ and noise $\epsilon \sim \mathcal{N}(\mathbf{0}, \mathbf{I})$:

$$x_t = \sqrt{\bar{\alpha}_t} x_0 + \sqrt{1 - \bar{\alpha}_t} \epsilon, \tag{3}$$

where $\bar{\alpha}_t := \prod_{s=1}^{t}(1 - \beta_s)$, and $\beta_s$ is the forward process variances fixed as constant in DDPM (Ho et al. (2020)). The loss of diffusion model is generally defined as follows:

$$\mathcal{L}_{diff} = ||\epsilon - \epsilon_\theta(x_t, t)||_2^2. \tag{4}$$

The noisy input $x_t$, accompanied by a time step embedding $t$, is input into a denoising autoencoder, specifically a U-Net network as in Ho et al. (2020), to estimate the noise component $\epsilon_\theta(x_t, t)$.

## 3.2 Proposed Distillation Framework

Our proposed framework is illustrated in Fig. 2. To reduce the activation peak memory, the initial pooling layer of the student is assigned a larger pooling stride. However, we still keep the same spatial dimensions of the activations to the final fully connected layer for the teacher and student. Thus, the student has fewer pooling layers but with a larger pooling stride at the initial pooling layer compared to the teacher.

Take an input image $x \in \mathbb{R}^{H \times W \times C}$ as the example. The teacher and the student network are divided into several stages by pooling layers as shown in Fig. 2. We assume that all pooling layers of the teacher have the same pooling stride $\rho$ for simplicity. The initial pooling layer of the student at stage 1 is assigned with a stride $\rho^k$. Hence, the student feature map $f_1^{\mathcal{S}} \in \mathbb{R}^{\frac{H}{\rho^k} \times \frac{W}{\rho^k} \times C_1}$ at the output of this pooling layer has the same spatial dimension as the teacher feature map $f_k^{\mathcal{T}} \in \mathbb{R}^{\frac{H}{\rho^k} \times \frac{W}{\rho^k} \times C_k}$ at the output of the $k$-th pooling layer. Then the matched feature maps $f_1^{\mathcal{S}}$ and $f_k^{\mathcal{T}}$ will be fed into the residual encoded distillation (RED) block illustrated in Fig. 3, to compute the distillation loss. The output $f_1^D$ is to be fed into the following layers of the student network.

Similarly, the feature map $f_2^{\mathcal{S}}$ of the student at stage 2 also has the same spatial dimension as the feature map $f_{k+1}^{\mathcal{T}}$ of the teacher at stage $k + 1$. They are fed into another RED block to calculate distillation loss

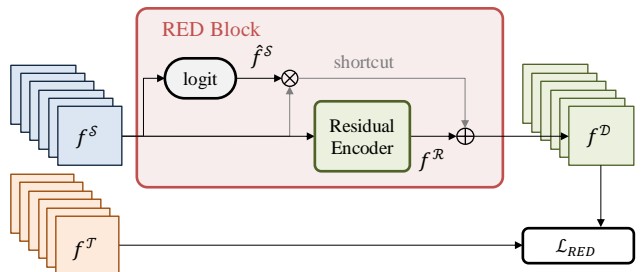

Figure 3: Residual Encoded Distillation (RED) Block. We use a logit module for the multiplicative gating mechanism and a residual encoder for additive residual learning.

and output $f_2^D$ to the following layers. This process is repeated until the last pooling layer $p_{\mathcal{T}}(\circ; \rho)$ at stage $n$ of the teacher. We assume the size of the output feature map $f_n^{\mathcal{T}}$ of the teacher is identical to the output feature map $f_i^{\mathcal{S}}$ of the student at stage $i$, where $i - 1 + k = n$. The following pooling layers of the student at stages $i + 1, i + 2, ..., n$ are all assigned with stride 1. Hence, these pooling layers do not change the spatial dimension and are similar to standard convolution. As a result, the final aggregated features of the student and the teacher have the same spatial dimensions. In Fig. 5, we show an example of ResNet18 with aggressive pooling and how to integrate our proposed RED blocks.

### 3.3 Residual Encoded Distillation Block

The proposed Residual Encoded Distillation (RED) Block, depicted in Fig. 3, serves as the central module of our framework. It is designed to ensure that the output of a pooling layer closely resembles the distribution of the input while preserving essential features at a reduced spatial dimension. To accomplish this, the RED Block is designed to be lightweight, enabling it to modify the feature space distribution of the student's pooling layer effectively. This allows the student model to learn the down-sampled features of the teacher model. Meanwhile, this block introduces non-linearity to the pooling layer, enabling the student's pooling layer to aggregate features like a standard pooling layer and adjust the feature distribution similar to a convolutional layer with an activation function. Specifically, we use a logit module for the multiplicative gating mechanism and a residual encoder for additive residual learning. The residual encoded distillation block could be formulated as follows:

$$
\begin{aligned}
f^{\mathcal{D}} &= f^{\mathcal{R}} + f^{\mathcal{S}} * \hat{f}^{\mathcal{S}}, & (5) \\
Residual\,Encoder &= ReLU6(BN(Conv_{3\times3}(\cdot))), & (6) \\
logit &= Sigmoid(BN(Conv_{1\times1}(\cdot))), & (7)
\end{aligned}
$$

where $f^{\mathcal{S}}$ is the feature map from the student model. The logit module consists of a $1 \times 1$ convolution layer, a batch norm layer, and a sigmoid activation function, generating element-wise weights $\hat{f}^{\mathcal{S}}$ like a gate to suppress non-significant components of $f^{\mathcal{S}}$. The residual encoder module consists of a $3\times3$ convolution layer, a batch norm layer, and a relu-6 activation function, yielding the residual item $f^{\mathcal{R}}$. We use relu-6 to bound activations to prevent exploding gradients. Besides, relu-6 is widely used in efficient neural network design (Sandler et al. (2018); Howard et al. (2019); Lin et al. (2020)) since it is particularly useful for fixed-point or low-precision inference in quantization. For kernel size in RED module, we choose $1 \times 1$ kernel for the logit module (LM), because LM is a gating mechanism to select critical components of the activations. We choose BatchNorm layer because the proposed framework is primarily used for the distillation of CNNs, for which BatchNorm is commonly used. We hypothesize that the output of the student pooling layers might lack some crucial information compared to the teacher's down-sampled features, which could be compensated by the residual item $f^{\mathcal{R}}$.

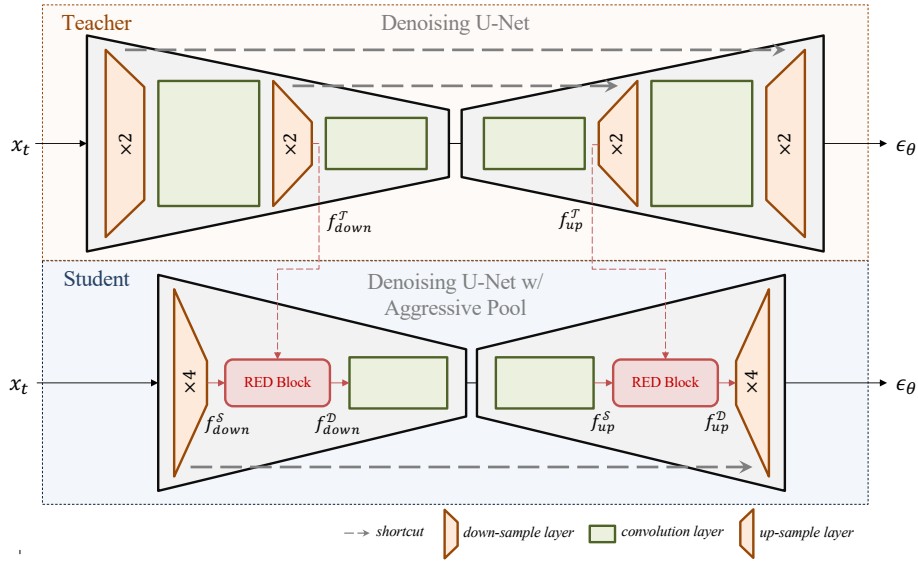

Figure 4: ReDistill for denosing network in DDPM (Ho et al. (2020)). We integrate RED blocks into the student model after the down-sampling layer in the encoder and before the up-sample layer in the decoder.

### 3.4 Loss Function

The RED loss first calculates the mean value alongside the channel dimension and then minimizes the cosine distance between the teacher's feature map $f^{\mathcal{T}}$ and the RED block's output $f^{\mathcal{D}}$, which is formulated as:

$$\mathcal{L}_{RED}(f^{\mathcal{T}}, f^{\mathcal{D}}) = d\{\frac{\sum_{c \in C_{\mathcal{T}}} f_c^{\mathcal{T}}}{|C_{\mathcal{T}}|}, \frac{\sum_{c \in C_{\mathcal{S}}} f_c^{\mathcal{D}}}{|C_{\mathcal{S}}|}\}, \tag{8}$$

where $f^{\mathcal{D}}$ is calculated by Equation 5, and $d$ denotes the distance measurement. $C_{\mathcal{T}}$ and $C_{\mathcal{S}}$ denote the channel dimension of teacher's feature map $f^{\mathcal{T}}$ and the RED block's output $f^{\mathcal{D}}$, respectively. The final loss function of the proposed method is as follows:

$$\mathcal{L} = \mathcal{L}_{task} + \sum_{i=1}^{I} \alpha \mathcal{L}_{RED_i}, \tag{9}$$

where $\mathcal{L}_{task}$ is the vanilla loss from the task, such as the Binary-Cross-Entropy (BCE) loss for image classification. $I$ denotes the number of RED blocks. $\alpha$ is an experimental hyper-parameter to scale RED loss.

### 3.5 Distillation for Diffusion Model

In this section, we introduce how to integrate the proposed distillation framework into a U-Net based denoising network described in DDPM (Ho et al. (2020)), as shown in Fig. 4. For convenience, let's assume the teacher model is a U-Net with two down-sample layers, each having a stride of 2, while the student is a U-Net with the aggressive pooling setting, i.e., it just has one down-sample layer with the stride 4. We use the output $f_{down}^{\mathcal{S}}$ from the down-sample layer of the student model and the output $f_{down}^{\mathcal{T}}$ from the second down-sample layer of the teacher model, as they share the same spatial dimension. $f_{down}^{\mathcal{S}}$ and $f_{down}^{\mathcal{T}}$ are input into a RED block incorporated into the student model, producing the output $f_{down}^{\mathcal{D}}$, which is then fed into the subsequent convolutional layers. Symmetrically, the student model has one up-sample layer with an expansion ratio of $\times 4$, while the teacher model has two up-sample layers, each with an expansion ratio of $\times 2$. The input $f_{up}^{\mathcal{S}}$ of the student's up-sample layer and the input $f_{up}^{\mathcal{T}}$ of the teacher's first up-sample are input into another RED block, producing the output $f_{up}^{\mathcal{D}}$. This output replaces the $f_{up}^{\mathcal{S}}$ as the new input of the student's up-sample layer. For DDPM distillation, the loss function is defined as Equation 9, while the $\mathcal{L}_{task}$ is replaced by $\mathcal{L}_{diff}$ defined in Equation 4.

# 4 Experiments

In Sections 4.1 and 4.2, datasets and implementation details for image classification and image generation tasks are introduced separately. In Sections 4.3 and 4.4, we conduct experiments on various vision tasks to illustrate the effectiveness of the proposed method with state-of-the-art distillation methods under different backbone architectures and datasets. We also compare the memory footprint of our method with the teacher and the student architectures, and deploy our method on the edge device. Results are reported in Section 4.5. At last, some ablation study related with module discussion, loss function, and distillation strategy are reported in Section 4.6.

## 4.1 Datasets

**Datasets for Image Classification** 1) STL10 (Coates et al. (2011)) contains 5K training images with 10 classes and 8K testing images of resolution $96 \times 96$ pixels. Specifically, we resize the image resolution to $128 \times 128$ pixels for aggressive pooling. 2) ImageNet (Russakovsky et al. (2015)) is a widely-used dataset of classification, which provides 1.2 million images for training and 50K images for validation over 1,000 classes. We keep the same resolution of $224 \times 224$ pixels as the origin for aggressive pooling.

**Datasets for Image Generation** 1) CIFAR-10 (Krizhevsky et al. (2009)) comprises 60,000 color images of 32x32 resolution across 10 classes, with each class containing 6,000 images. The dataset is divided into 50,000 training images and 10,000 test images. We keep the original resolution of $32 \times 32$ in our experiments. 2) Celeb-A (Liu et al. (2015)) is a large-scale face attributes dataset containing over 200,000 celebrity images, each annotated with 40 attributes. We use the resized resolution of $64 \times 64$, which is widely used in diffusion-based methods (Song et al. (2020); Bao et al. (2023)) in our experiments.

## 4.2 Implementation Details and Baselines

**Details for Image Classification** Unlike traditional distillation tasks, we only modify the pooling layer strides instead of the depth and width of the network to get the student model, which is called the aggressive pooling setting. The advantage of the aggressive pooling setting is to reduce the peak memory and also reduce the computational complexity and inference time of the network. Specifically, we increase the first pooling layer stride $\times 2 \sim \times 8$ times and adjust the last several pooling layer strides to ensure the final output of the student model has the same information density as the teacher model. All experiments are implemented in Pytorch and evaluated on 4 NVIDIA A100 GPUs.

On STL10 dataset, we experiment with three representative and widely-used network architectures, including MobileNetV2 (Sandler et al. (2018)), MobileNetV3 (Howard et al. (2019)), and ResNext (Xie et al. (2017)). The proposed method is compared with several distillation methods (Hinton et al. (2015); Adriana et al. (2015); Zagoruyko & Komodakis (2016); Tung & Mori (2019); Ahn et al. (2019); Park et al. (2019); Heo et al. (2019); Kim et al. (2018); Huang & Wang (2017)). Specifically, the student architecture is trained from scratch as being distilled from pre-trained teacher architecture by different methods for 300 epochs. The batch size is set to 8 and the dropout rate is set to 0.2. The SGD with momentum equal to 0.9 is used as the optimizer. The initial learning rate is set to 0.01, which is reduced by a factor 0.2 at the $180^{th}$, $240^{th}$ and $270^{th}$ epoch, respectively. The $\alpha$ in Equation 9 is set to 50.

On ImageNet dataset, we experiment on the MobileNetV2 (Sandler et al. (2018)), and ResNet (He et al. (2016)), which are widely used in distillation benchmarks (Jin et al. (2023); Chen et al. (2021); Tian et al. (2020)). The proposed method is compared with response-based methods like KD (Hinton et al. (2015)), DKD (Zhao et al. (2022)), MLLD (Jin et al. (2023)), LSKD (Sun et al. (2024)), and feature-based methods like FitNet (Adriana et al. (2015)), RKD (Park et al. (2019)), ReviewKD (Chen et al. (2021)), CRD (Tian et al. (2020)), and OFAKD (Hao et al. (2023)). All these methods are widely used in knowledge distillation and, to the best of our knowledge, yield SOTA performance. We use the same experiment settings as (Jin et al. (2023)) but keep training for 300 epochs and decay the learning rate at the $180^{th}$, $240^{th}$ and $270^{th}$ epoch with factor 0.1, since the student architectures with the aggressive pooling setting generally require more epochs to converge. The $\alpha$ in Equation 9 is set to be 1.

**Details for Image Generation** For the teacher model, we keep the same experiment settings as DDPM (Ho et al. (2020)) with applying $T = 1000$, $\beta_1 = 10^{-4}$, $\beta_T = 0.02$, and the U-Net backbone with 4 different feature map resolutions ($32 \times 32$ to $4 \times 4$ for CIFAR-10, while $64 \times 64$ to $8 \times 8$ for Celeb-A). For the student model, we increase the first pooling layer stride of the U-Net backbone $\times 2$ times while adjusting the last pooling layer stride to keep the same latent feature resolution. The same stride modification is symmetrically applied to the up-sample layers of the U-Net, and thus with 3 different feature map resolutions ($16 \times 16$ to $4 \times 4$ for CIFAR-10, while $32 \times 32$ to $8 \times 8$ for Celeb-A). For our method, the RED blocks are inserted after not only the down-sample layers but also the up-sample layers. For CIFAR-10 dataset, we train all models for 1000K iterations and sample 50K images for FID (Heusel et al. (2017)) & IS (Salimans et al. (2016)) evaluation. For Celeb-A dataset, we train all models with 250K iterations and sample 50K images for FID (Heusel et al. (2017)) & IS (Salimans et al. (2016)) evaluation. All experiments are implemented in Pytorch and evaluated on an NVIDIA 4090 GPU.

**Details for Theoretical Peak Memory** For the theoretical peak memory analysis, we follow previous work, MCUNet (Lin et al. (2020; 2021; 2022)), for calculating peak memory. Generally, it traces the memory consumption for each layer. For the standard convolutional layer, the maxpool layer, or the average pool layer, the memory consumption is the summation of the input activation memory size and output activation memory size. For a group convolutional layer, the memory consumption is the summation of the input activation memory size, the output activation memory size, and a buffer with size equal to one channel convolutional kernel in the group. For the residual connection in the network, the residual activations memory size will be added into memory tracing until the residual item is added with the output activations.

**Details for Aggressive Pooling Setting.** To explain how exactly the image classification models are configured, we list the network config details for the 'T: ResNet18' and 'S: ResNet18×4' distillation pair that we used in the third column of Table 1. As shown in Fig. 5, we aggressively increase the stride of the first downsampling layer 'Conv 1' from 2 to 8 following the aggressive pooling setting, while setting the stride to 1 of maxpool layer and the last downsampling layer, i.e., the first conv layer in 'Stage 4', for keeping the same activation size for average pool and fc layer. Then the student network possesses only three downsampling layers less than five downsapling layers of the teacher network. We integrate the proposed RED block after each downsampling layer of the student in our ReDistill framework to improve the student's performance while maintaining low peak memory.

## 4.3 CNN-based Image Classification

Table 1 shows the results on the ImageNet (Russakovsky et al. (2015)) dataset, with the setting that the teacher model and student model are in identical architecture families or different architectures. '$\times n$' denotes that we increase the $1^{st}$ pooling layer stride of this architecture with $n$ times, and the best results are highlighted in boldface. Our method achieves the best performance compared with different response-based and feature-based distillation methods, no matter for identical architecture family knowledge distillation, as shown in the $3^{rd}$ and $4^{th}$ columns of Table 1, or different architecture knowledge distillation, as shown in the $5^{th}$ column of Table 1.

We find some distillation methods perform even worse than the student model itself without any distillation since these methods are not specially designed for the aggressive pooling setting and would be sensitive to the resolution of the feature maps, like FitNet, or require multi-scale feature maps, like RKD. Generally, conventional feature-based methods match the activations of the teacher model and student model stage by stage. In each stage, these activations possess the same resolution. However, in our aggressive pooling setting, the resolution of student activations and teacher activations are mismatched in each stage, and thus, traditional distillation methods are not guaranteed to be positively effective in knowledge distillation with aggressive pooling. This observation also illustrates the necessity of the proposed ReDistill framework.

Table 2 shows the classification results on STL10 dataset with the setting that the teacher model and student model are in identical architecture family. Same as ImageNet, '$\times n$' denotes we increase the $1^{st}$ pooling layer stride of this architecture with $n$ times, and the best results are highlighted in boldface. Some KD methods in Table 1 require a customized dataloader with a contrastive version of the input data, like CRD Tian et al.

| Network Config | T: ResNet18 | | S: ResNet18×4 | | S + RED (ours) | |
|---|---|---|---|---|---|---|
| | layer config | activation size | layer config | activation size | layer config | activation size |
| **Input** | - | 224×224 | - | 224×224 | - | 224×224 |
| **Conv 1** | *conv* 7×7: *c_64, s_2* | 112×112 | *conv* 7×7: *c_64, s_8* | 28×28 | *conv* 7×7: *c_64, s_8* **+ RED block** | 28×28 |
| **Max Pool + Stage 1** | *maxpool* 3×3: *s_2* $\begin{bmatrix} conv\ 3×3:c\_64, s\_1 \\ conv\ 3×3:c\_64, s\_1 \end{bmatrix}$ $\begin{bmatrix} conv\ 3×3:c\_64, s\_1 \\ conv\ 3×3:c\_64, s\_1 \end{bmatrix}$ | 56×56 | *maxpool* 3×3: *s_1* $\begin{bmatrix} conv\ 3×3:c\_64, s\_1 \\ conv\ 3×3:c\_64, s\_1 \end{bmatrix}$ $\begin{bmatrix} conv\ 3×3:c\_64, s\_1 \\ conv\ 3×3:c\_64, s\_1 \end{bmatrix}$ | 28×28 | *maxpool* 3×3: *s_1* $\begin{bmatrix} conv\ 3×3:c\_64, s\_1 \\ conv\ 3×3:c\_64, s\_1 \end{bmatrix}$ $\begin{bmatrix} conv\ 3×3:c\_64, s\_1 \\ conv\ 3×3:c\_64, s\_1 \end{bmatrix}$ | 28×28 |
| **Stage 2** | $\begin{bmatrix} conv\ 3×3:c\_128, s\_2 \\ conv\ 3×3:c\_128, s\_1 \end{bmatrix}$ $\begin{bmatrix} conv\ 3×3:c\_128, s\_1 \\ conv\ 3×3:c\_128, s\_1 \end{bmatrix}$ | 28×28 | $\begin{bmatrix} conv\ 3×3:c\_128, s\_2 \\ conv\ 3×3:c\_128, s\_1 \end{bmatrix}$ $\begin{bmatrix} conv\ 3×3:c\_128, s\_1 \\ conv\ 3×3:c\_128, s\_1 \end{bmatrix}$ | 14×14 | $\begin{bmatrix} conv\ 3×3:c\_128, s\_2 \\ +RED\ block \\ conv\ 3×3:c\_128, s\_1 \end{bmatrix}$ $\begin{bmatrix} conv\ 3×3:c\_128, s\_1 \\ conv\ 3×3:c\_128, s\_1 \end{bmatrix}$ | 14×14 |
| **Stage 3** | $\begin{bmatrix} conv\ 3×3:c\_256, s\_2 \\ conv\ 3×3:c\_256, s\_1 \end{bmatrix}$ $\begin{bmatrix} conv\ 3×3:c\_256\ s\_1 \\ conv\ 3×3:c\_256, s\_1 \end{bmatrix}$ | 14×14 | $\begin{bmatrix} conv\ 3×3:c\_256, s\_2 \\ conv\ 3×3:c\_256, s\_1 \end{bmatrix}$ $\begin{bmatrix} conv\ 3×3:c\_256\ s\_1 \\ conv\ 3×3:c\_256, s\_1 \end{bmatrix}$ | 7×7 | $\begin{bmatrix} conv\ 3×3:c\_256, s\_2 \\ + RED\ block \\ conv\ 3×3:c\_256, s\_1 \end{bmatrix}$ $\begin{bmatrix} conv\ 3×3:c\_256\ s\_1 \\ conv\ 3×3:c\_256, s\_1 \end{bmatrix}$ | 7×7 |
| **Stage 4** | $\begin{bmatrix} conv\ 3×3:c\_512, s\_2 \\ conv\ 3×3:c\_512, s\_1 \end{bmatrix}$ $\begin{bmatrix} conv\ 3×3:c\_512\ s\_1 \\ conv\ 3×3:c\_512, s\_1 \end{bmatrix}$ | 7×7 | $\begin{bmatrix} conv\ 3×3:c\_512, s\_1 \\ conv\ 3×3:c\_512, s\_1 \end{bmatrix}$ $\begin{bmatrix} conv\ 3×3:c\_512\ s\_1 \\ conv\ 3×3:c\_512, s\_1 \end{bmatrix}$ | 7×7 | $\begin{bmatrix} conv\ 3×3:c\_512, s\_1 \\ conv\ 3×3:c\_512, s\_1 \end{bmatrix}$ $\begin{bmatrix} conv\ 3×3:c\_512\ s\_1 \\ conv\ 3×3:c\_512, s\_1 \end{bmatrix}$ | 7×7 |
| **Avg Pool + FC** | average pool, 1000-d fc, softmax | 1×1 | average pool, 1000-d fc, softmax | 1×1 | average pool, 1000-d fc, softmax | 1×1 |

Figure 5: Example of our proposed Aggressive Pooling Setting with ResNet18. We highlight all downsampling layers (either conv layer or maxpool layer with the stride larger than 1) in red color. For example, in the 'Conv 1' cell of 'T: ResNet18', 'conv $7 \times 7 : c\_64, s\_2$' denotes this convolution layer is with kernel size $7 \times 7$, number of channels 64, and stride 2. In the following row, 'maxpool $3 \times 3 : s\_2$' denotes the max pool layer with kernel size $3 \times 3$ and stride 2. In the student network 'S: ResNet18×4', we increase the stride of the first downsampling layer 'Conv 1' to $4\times$ from $s\_2$ to $s\_8$, while setting the stride to 1 of the maxpool layer and the last downsampling layer (i.e., the first conv layer in Stage 4), in order to get the same activation size before average pool and fc layer.

Table 1: Top-1 accuracy (%) on ImageNet. The $3^{rd}$ and $4^{th}$ columns show the results with identical teacher and student architecture families, while the $5^{th}$ column shows the results with different teacher and student architectures.

| Method | Teacher | ResNet18 | ResNet50 | ResNet152 |
|---|---|---|---|---|
| | Top1 Acc. (%) | 69.75 | 76.13 | 78.32 |
| | Student | ResNet18×4 | ResNet50×4 | MbNetV2×2 |
| | Top1 Acc. (%) | 61.79 | 69.50 | 62.65 |
| Response | KD (Hinton et al. (2015)) | 63.63 | 70.60 | 62.85 |
| | DKD (Zhao et al. (2022)) | 63.22 | - | 66.27 |
| | MLLD (Jin et al. (2023)) | 64.66 | 70.77 | 68.36 |
| | LSKD (Sun et al. (2024)) | 63.81 | 72.28 | 66.21 |
| Feature | FitNet (Adriana et al. (2015)) | 62.13 | 71.77 | 60.79 |
| | RKD (Park et al. (2019)) | 61.49 | 66.88 | - |
| | ReviewKD (Chen et al. (2021)) | 63.30 | 70.22 | 63.07 |
| | CRD (Tian et al. (2020)) | 64.01 | 71.07 | 65.60 |
| | OFAKD (Hao et al. (2023)) | 64.83 | - | 68.49 |
| | **RED (ours)** | **65.23** | **73.23** | **68.89** |

'-' denotes that we do not get reasonable results for the student architecture with the distillation method under the aggressive pooling setting.

(2020), MLLD Jin et al. (2023). However, they don't provide the corresponding dataloader for STL10 dataset in their original code implementations. Thus, we compare our method on STL10 dataset with some feasible KD methods in Table 1, like KD, FitNet and RKD, and also compare with some other commonly-used KD methods, like AT, VID, NST, etc. Under the identical architecture family setting, our method performs the best among all state-of-the-art distillation methods. We also measure the average distillation time for different distillation methods, as shown in the last column of Table 2. Our method achieves similar time consumption compared to most distillation methods. For different architecture knowledge distillation, as shown in Table 3, our method achieves better performance compared with other distillation methods as well.

## 4.4 Image Generation

**Quantitative Results** Table 4 shows the results on CIFAR10 (Krizhevsky et al. (2009)) dataset and Celeb-A (Liu et al. (2015)) dataset. Our method reduces the fidelity degradation of the student model with a first pooling stride that is twice that of the teacher model. Specifically, our method achieves 3.43 lower FID and 0.51 higher IS score than the student model on CIFAR10 dataset, while 1.6 lower FID and 0.11 higher IS score than the student model on Celeb-A, respectively. We also compare to a simple distillation method, which we termed MSE-Distill-DDPM, by directly matching the final output activations of the teacher model

Table 4: Results on U-Net (Ronneberger et al. (2015)) based DDPM (Ho et al. (2020)).

| Dataset | Method | IS↑ | FID↓ |
|---------|--------|-----|------|
| CIFAR10 | T: U-Net w/ DDPM | **9.70 ±0.13** | **3.75** |
| | S: U-Net×2 w/ DDPM | 9.12 ± 0.14 | 14.28 |
| | S + MSE-Distill | 9.62 ± 0.11 | 12.04 |
| | **S w/ RED (ours)** | **9.63 ± 0.10** | **10.85** |
| Celeb-A | T: U-Net w/ DDPM | **2.97 ± 0.04** | **19.61** |
| | S: U-Net×2 w/ DDPM | 2.77 ± 0.02 | 23.03 |
| | S + MSE-Distill | 2.71 ± 0.02 | 22.34 |
| | **S w/ RED (ours)** | **2.88 ± 0.03** | **21.16** |

and student model using MSE loss. On CIFAR10 dataset, the student with MSE-Distill achieves 12.04 FID and 9.62 IS score, while our method achieves a better 10.85 FID and 9.63 IS score. Due to implementation specifics, the models are not augmented with EMA (Hunter (1986)), leading to results slightly different from those in the original paper.

**Visualization** To further illustrate how the proposed method improves the fidelity of image generation, we visualize some samples generated by the teacher model, the student model, and our method. Specifically, we utilize the same noise item $\epsilon_t$ in each time step for all these three models. In this way, the generated images are expected to be visually similar if the two models have comparable capabilities. As shown in Fig. 6, generally, the images generated by our method are semantically closer to those generated by the

Table 2: Top-1 accuracy (%) on STL10 with identical teacher and student architecture family.

| Method | T: MbNetV2 85.34 S: MbNetV2×4 76.18 | T: MbNetV3-Small 83.74 S: MbNetV3-S×4 71.27 | T: ResNext18 85.12 S: ResNext18×4 79.07 | Distill Time (s/epoch) |
|--------|------|------|------|------|
| KD (Hinton et al. (2015)) | 79.26 | 71.56 | 81.27 | **3.23** |
| FitNet (Adriana et al. (2015)) | 78.55 | 71.93 | 80.21 | 3.28 |
| AT (Zagoruyko & Komodakis (2016)) | 81.35 | 75.88 | 82.90 | 3.33 |
| SP (Tung & Mori (2019)) | - | 72.42 | 77.59 | 3.31 |
| VID (Ahn et al. (2019)) | 77.53 | 71.83 | 76.99 | 3.40 |
| RKD (Park et al. (2019)) | 78.00 | 67.27 | 75.80 | 3.33 |
| AB (Heo et al. (2019)) | 80.79 | 72.94 | 81.42 | 3.69 |
| FT (Kim et al. (2018)) | 74.74 | 72.11 | 80.64 | 3.35 |
| NST (Huang & Wang (2017)) | 82.66 | 76.11 | 81.89 | 9.33 |
| **RED (ours)** | **83.97** | **77.31** | **84.80** | 3.88 |

'-' denotes that we do not get reasonable results for the student architecture with the distillation method under the aggressive pooling setting.

Table 3: Top-1 accuracy (%) on STL10 with different teacher and student architecture.

| Method | T: ResNext18 85.12 S: MbNetV2×4 76.18 | T: ResNext18 85.12 S: MbNetV3-Small×4 71.27 | T: MbNetV2 85.34 S: MbNetV3-Small×4 71.27 |
|---|---|---|---|
| KD (Hinton et al. (2015)) | 78.14 | 69.98 | 70.95 |
| FitNet (Adriana et al. (2015)) | 80.46 | 73.23 | 72.88 |
| AT (Zagoruyko & Komodakis (2016)) | 80.63 | 74.58 | 73.39 |
| SP (Tung & Mori (2019)) | 67.56 | 63.91 | - |
| VID (Ahn et al. (2019)) | 74.40 | 69.19 | 71.69 |
| RKD (Park et al. (2019)) | 70.63 | 72.10 | 68.78 |
| AB (Heo et al. (2019)) | 81.46 | 73.98 | 75.20 |
| FT (Kim et al. (2018)) | 77.33 | 69.79 | 66.85 |
| NST (Huang & Wang (2017)) | 79.09 | 65.04 | 73.26 |
| **RED (ours)** | **83.23** | **77.15** | **77.19** |

'-' denotes that we don't get reasonable results for the student architecture with the distillation method under the aggressive pooling setting.

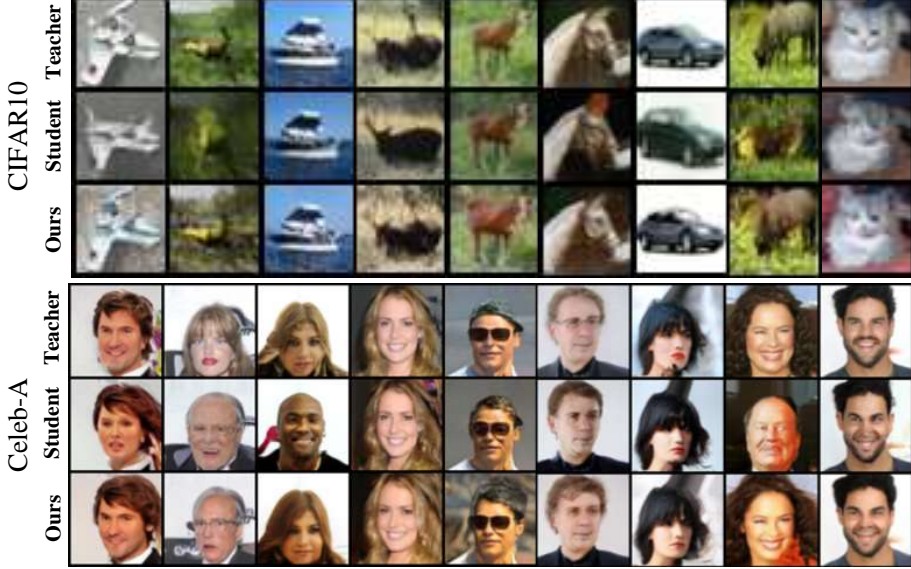

Figure 6: Generated images on CIFAR10 and Celeb-A with the same noise item $\epsilon_t$ for all models. Generally, our results are semantically closer to the teacher's.

teacher model, although in some cases, they are closer to those generated by the student model. Due to the high capability of the teacher model, our method, as an intermediate model from the teacher to the student, achieves higher fidelity than the student model.

## 4.5 Memory Footprint and Edge Device Deployment

**Memory Footprint** To intuitively demonstrate the peak memory reduction enabled by our proposed method, we trace the memory footprint in layer-wise for all identical architecture family teacher-student pairs, as shown in Fig. 7. Compared to CNN models with vanilla pooling, our method with aggressive pooling settings achieves significantly lower memory consumption, particularly in the initial layers where peak memory usage occurs. For DDPM, which utilizes a U-Net architecture with both down-sampling and up-sampling layers, our method reduces memory consumption in both the initial layers and the final layers where peak memory usage occurs.

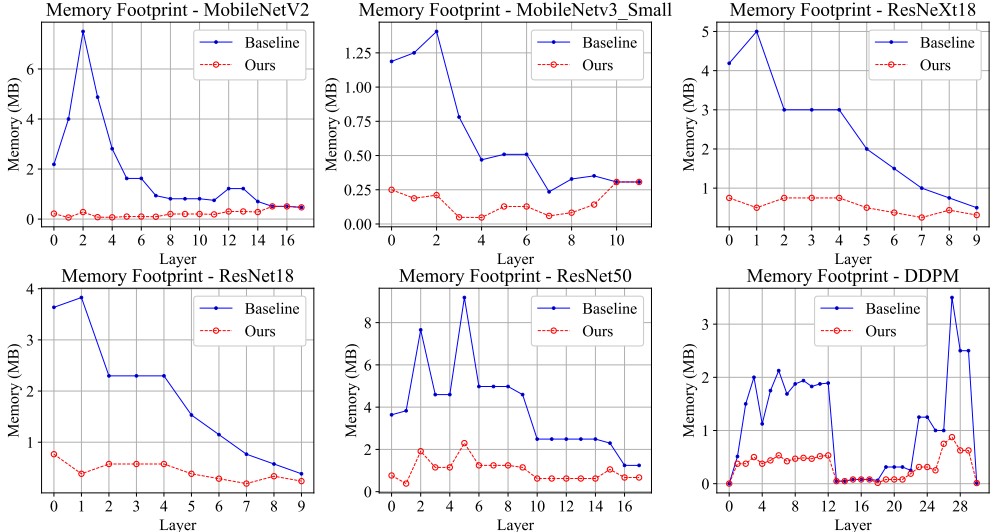

Figure 7: Memory footprint for each layer in teacher and student networks (top-all: STL10 classification task; bottom-left, bottom-mid: ImageNet classification task; bottom-right: Celeb-A generation task). The student model is the teacher model's modified version of being assigned with the aggressive pooling strategy and enhanced with our proposed RED blocks, representing our method.

**Edge Device Deployment**    In Table 5, we measure theoretical peak memory consumption, actual peak memory consumption, and other efficiency related metrics like model size, maximum power, and latency, for all the models we use in STL10 & ImageNet classification and image generation tasks. Specifically, the $1^{st}$-$9^{th}$ rows are for models we use in STL10 classification, and the $10^{th}$-$18^{th}$ row are for models we use in ImageNet classification, while the last 3 rows are for models we use in image generation task. We estimate the theoretical peak memory by summing the size of the input & output allocation for each operation and assume a batch size of 1 for theoretical estimation to emulate most inference use-cases, similar to (Lin et al. (2021); Chowdhery et al. (2019)). Our method achieves $3.9 \times \sim 14.7\times$ reduction in theoretical peak memory for image classification and $4\times$ reduction in theoretical peak memory for image generation, as shown in Table 5. In addition to the theoretical peak memory, we also measure the actual peak memory consumed on an NVIDIA Jetson TX2 device. Moreover, we measure the maximum GPU power (to ensure we satisfy the power budget of edge devices), model size, and latency incurred by the baseline teacher, student, and our RED models on the same edge GPU. Our models yield similar peak memory as the student models (due to similar levels of aggressive striding) and $\sim 2\times \sim 3.2\times$ lower peak memory for image classification and $\sim 2\times$ lower peak memory for image generation tasks compared to the teacher models. Note that our theoretical and measured peak memory reduction factors are different due to varying device setups and buffer allocations. However, our models incur worse latency and power compared to the student models due to the additional RED blocks and improved latency compared to the teacher models.

### 4.6    Ablation Study

**Module Discussion**    As shown in Table 6, we conduct the ablation study on STL10 dataset with the MobileNetV3-Small backbone by removing the logit module, residual encoder, shortcut, and the whole RED block, respectively. The performance of our proposed model degrades slightly without the logit module but seriously degrades without the residual encoder, which illustrates that the residual encoder plays a more significant role in the RED block. Without RED blocks, the student model has poor performance when only applying RED loss on the student activation maps and teacher activation maps, which illustrates the necessity to integrate RED blocks into the student model. Naively stacking the logit module and residual encoder, i.e., without Shortcut, performs even worse than without RED blocks, illustrating the effectiveness of our designed shortcuts in the logit module and residual encoder. We also conduct the ablation study on

Table 5: **Edge Device Deployment.** We analyze the theoretical Peak Memory (T-PkMem), and measure actual GPU Peak Memory (A-PkMem), Model Size (MS), Maximum Power (MxP), and Latency for models on an NVIDIA Jetson TX2 device. For theoretical Peak Memory analysis, the batch size is assumed to be 1. For each actual measurement, the batch size (BS) is reported in the table and set to the maximum load allowed by the device, which is restricted by the teacher. The $1^{st}$-$9^{th}$ rows are for models we use in STL10 classification, and the $10^{th}$-$18^{th}$ row are for models we use in ImageNet classification, while the last 3 rows are for models we use in image generation task. The overheads of RED modules on top of student networks is minimal in particular for cases with more aggressive pooling, e.g., distillation of MbNetV2 with ×8 pooling.

| Model | Theoretical Analysis | Actual Measurement | | | | |
| | T-PkMem (MB) | BS | A-PkMem (GB) | MS (MB) | MxP (mW) | Latency (ms) |
|---|---|---|---|---|---|---|
| T: MbNetV2 | 7.50 | | 2.30 | **13.50** | 1830 | 221.53 ± 2.34 |
| S: MbNetV2×8 | **0.51** | 20 | **0.78** | **13.50** | **458** | **49.51 ± 1.98** |
| **S w/ RED (ours)** | **0.51** | | **0.78** | 14.25 | **534** | **55.43 ± 1.80** |
| T: MbNetV3-Small | 1.41 | | 2.30 | **9.75** | 1526 | 195.63 ± 1.09 |
| S: MbNetV3-Small×4 | **0.31** | 50 | **1.10** | **9.75** | **611** | **69.76 ± 1.55** |
| **S w/ RED (ours)** | **0.31** | | **1.10** | 10.18 | **687** | **83.74 ± 1.40** |
| T: ResNext18 | 5.00 | | 2.40 | **21.47** | 3276 | 448.38 ± 6.97 |
| S: ResNext18×4 | **0.75** | 50 | **1.10** | **21.47** | **1450** | **173.23 ± 1.92** |
| **S w/ RED (ours)** | **0.75** | | **1.10** | 25.56 | **2058** | **275.02 ± 3.25** |
| T: ResNet18 | 3.83 | | 2.70 | **44.63** | 3273 | 395.42 ± 4.21 |
| S: ResNet18×4 | **0.77** | 50 | **1.20** | **44.63** | **1297** | **151.59 ± 1.50** |
| **S w/ RED (ours)** | **0.77** | | **1.20** | 48.72 | **1830** | **208.43 ± 1.53** |
| T: ResNet50 | 9.19 | | 2.00 | **97.70** | 2591 | 293.81 ± 1.09 |
| S: ResNet50×4 | **2.30** | 10 | **1.00** | **97.70** | **1068** | **125.49 ± 1.17** |
| **S w/ RED (ours)** | **2.30** | | **1.00** | 160.44 | **2136** | **246.68 ± 1.73** |
| T: ResNet152 | 9.19 | | 2.80 | 230.20 | 3880 | 612.03 ± 3.55 |
| S: MobileNetV2×2 | **1.15** | 8 | **0.71** | **16.23** | **305** | **23.98 ± 0.84** |
| **S w/ RED (ours)** | **1.15** | | **0.71** | 32.73 | **611** | **72.65 ± 1.01** |
| T: UNet w/ DDPM | 3.50 | | 2.90 | **133.09** | 4257 | 684.31 ± 5.39 |
| S: UNet×2 w/ DDPM | **0.88** | 30 | **1.30** | **133.09** | **1983** | **249.92 ± 3.76** |
| **S w/ RED (ours)** | **0.88** | | **1.30** | 149.92 | **2362** | **304.97 ± 1.87** |

Table 6: Module Discussion on STL10 dataset with MbNetV3-Small×4 with $\alpha$=50, and cosine distance for the RED loss. LM denotes Logit Module, and RE denotes Residual Encoder. ks denotes the kernel size of the convolution layer in RE.

| Method | RED block | | | | Top1 Acc. (%) |
| | LM | RE | Shortcut | ks | |
|---|---|---|---|---|---|
| w/o LM | - | ✓ | ✓ | 3 | 76.66 |
| w/o RE | ✓ | - | ✓ | - | 72.30 |
| w/o Shortcut | ✓ | ✓ | - | 3 | 52.42 |
| w/o RED block | - | - | - | - | 69.31 |
| RE w/ ks=1 | ✓ | ✓ | ✓ | 1 | 73.86 |
| RE w/ ks=5 | ✓ | ✓ | ✓ | 5 | 76.26 |
| **RED (ours)** | ✓ | ✓ | ✓ | 3 | **77.31** |

the kernel size of the residual module of our RED block. It shows that the proposed method achieves the best performance with kernel size 3.

**Distillation Strategy** We conduct the ablation study to compare different strategies during distillation, including the feature alignment strategy and pooling stride of the initial layer. Specifically, we train the ResNext18 network on the STL10 dataset with three different strategies while enlarging the stride of the first pooling layer to ×2, ×4, and ×8. The results are shown in Table 7. 'RED w/o Distillation' is the baseline of

Table 7: Ablation study of distillation strategies on STL10 dataset with ResNext18.

| Backbone: ResNext18 | Top1 Acc. (%) | | |
|---|---|---|---|
| $1^{st}$ Pooling Stride | ×2 | ×4 | ×8 |
| RED w/o Distillation | 81.43 | 79.07 | 73.16 |
| RED w/ Stage-Align | 80.59 | 81.99 | 74.00 |
| **RED w/ Pooling-Align (ours)** | **85.51** | **84.80** | **77.48** |

training an aggressive pooled ResNext18 without distillation. 'RED w/ Stage-Align' applies the traditional stage-based feature alignment between the student and the teacher activations for distillation. 'RED w/ Pooling-Align' is the proposed pooling-based feature alignment strategy. Compared to traditional stage-based alignment, the proposed pooling-based alignment performs much better for the aggressive pooling strategy.

**Loss Function**  We also conduct the ablation study for the proposed RED loss function, an equally important portion of distillation. The results are based on ResNext18 backbone and STL10 dataset, as shown in Table 8. Without the RED loss, i.e. setting $\alpha$ to 0 and then the loss function $\mathcal{L} = \mathcal{L}_{task}$, the student model is just integrated with RED blocks but without distillation. In this circumstance, the poor performance of the student model illustrates the need to apply the loss of RED. The proposed method performs worse when applying the Euclidean distance instead of the cosine distance to calculate the RED loss. Averaging along the channel dimension with different $C_T$ and $C_S$ in Equation 8 might mitigate the absolute difference in Euclidean space, while cosine distance measures the angle between two vectors, preserving their relative difference. By

Table 8: Loss Function Discussion on STL10 dataset with ResNext18×4. We conduct ablation study on $\alpha$ and distance measurement for $\mathcal{L}_{RED}$. We also compare our method with the student integrated with the proposed RED block but distilled only by $\mathcal{L}_{KD}$ and distilled by both our $\mathcal{L}_{RED}$ and $\mathcal{L}_{KD}$.

| Method | $\mathcal{L}_{RED}$ | | $\mathcal{L}_{KD}$ | Top1 Acc. (%) |
|---|---|---|---|---|
| | $\alpha$ | Distance | | |
| **RED (ours)** | 0 | - | - | 82.26 |
| | 50 | Euclidean | - | 82.66 |
| | 1 | Cosine | - | 82.04 |
| | 5 | Cosine | - | 82.86 |
| | 10 | Cosine | - | 83.41 |
| | 50 | Cosine | - | 84.80 |
| | 100 | Cosine | - | **85.10** |
| | 200 | Cosine | - | 84.95 |
| **RED$_{w/o\mathcal{L}_{RED}}$+KD** | 0 | - | ✓ | 81.94 |
| **RED (ours)+KD** | 50 | Cosine | ✓ | 84.82 |

adjusting $\alpha$ from 200 to 1, the performance degradation shows that the proposed method is sensitive to the hyper-parameter $\alpha$ in Equation 9. Besides, we evaluate the proposed method's performance when incorporated with other knowledge distillation methods, such as KD (Hinton et al. (2015)). Specifically, we first evaluate the student model integrated with the proposed RED blocks by applying only KD loss, which is denoted as 'RED$_{w/o\mathcal{L}_{RED}}$+KD'. Then we apply both RED loss and KD loss into the same student model, denoted as 'RED (ours)+KD'. Performance improvement illustrates the effectiveness of the proposed RED loss.

**Distillation w/o Aggressive Pooling**  In addition to aggressive pooling, we conduct experiments to compare several distillation methods from the teacher network ResNext50 to the student network MobileNetV3-Small without aggressive pooling on STL10 dataset. As shown in Table 9, our method still achieves the best performance, illustrating the generalization ability of the proposed framework.

**Comparison to Non-distillation Method**  We compare our methods to quantization-aware-training (QAT) method by using the pytorch native architecture optimization toolkit TorchAO (torchao maintainers & contributors (2024)) and a 6 bits QAT method, INQ (Aojun Zhou (2017)). We also deploy Intel Neural Compressor (INC) (Cheng et al. (2023)) to conduct int8 fully precision and

Table 9: Distillation w/o Aggressive Pooling Setting.

| Method | Top1 Acc. (%) |
|---|---|
| T: ResNext50 | 83.54 |
| S: MbNetV3-Small | 72.62 |
| KD | 77.06 |
| FitNet | 76.35 |
| AT | 76.76 |
| VID | 74.83 |
| RKD | 73.96 |
| AB | 77.50 |
| NST | 73.78 |
| RED (ours) | **79.71** |

int8/fp32 mixed precision post-training-quantization (PTQ) methods. As shown in Table 10, the ResNext18 quantized by INC to int8 achieves 20.12% accuracy and 1.25MB peak memory. The ResNext18 quantized by INC to mixed precision achieves 84.18% accuracy and 4.19 MB peak memory. The ResNext18 quantized by INQ to 6 bits achieves 84.07% accuracy and 5.00 MB peak memory. Our distillation method achieves the optimal trade-off between accuracy and peak memory, with 84.80% accuracy and 0.75 MB peak memory. Due to the extra RED blocks, our method possesses the higher model size of 25.557 MB. However, our method is orthogonal to these quantization methods and thus can be applied with the quantization method to further optimize the student model's overhead. As shown in the last row of Table 10, the student model ResNext18×4 distilled by our method and quantized by INC to mixed precision achieves 84.78% accuracy, 0.44 MB peak memory, and 5.621 MB model size.

Table 10: Comparison w/ Non-distillation Method.

| Method | Top1 Acc.(%) | T-PkMem (MB) | Model Size (MB) |
|---|---|---|---|
| T: ResNext18 - fp32 | **85.12** | 5.00 | 21.47 |
| w/ TorchAO - dynamic, QAT | 83.37 | 5.00 | 19.513 |
| w/ INC - int8, PTQ | 20.12 | 1.25 | 4.960 |
| w/ INC - mixed prec., PTQ | 84.18 | 4.19 | 5.015 |
| w/ INQ - 6 bits, QAT | 84.07 | 5.00 | **4.026** |
| S: ResNext18×4 - fp32 | 79.07 | 0.75 | 21.47 |
| w/ INC - mixed prec., PTQ | 78.50 | **0.44** | 5.015 |
| S + **RED (ours)** - fp32 | **84.80** | 0.75 | 25.557 |
| w/ INC - mixed prec., PTQ | 84.78 | **0.44** | 5.621 |

## 5 Conclusions and Future Work

We propose ReDistill, a novel residual encoded distillation method to reduce the peak memory of convolutional neural networks during inference. Our method enables the deployment of these networks in edge devices, such as micro-controllers with tight memory budgets, while accommodating high-resolution images necessary for intricate vision tasks. The reduced peak memory can also enable these networks to be implemented with recently proposed in-sensor computing systems (Datta et al. (2022; 2023)), thereby significantly reducing the bandwidth between the image sensor and the back-end processing unit. Our method is based on a teacher-student distillation framework, where the student network using aggressive pooling with reduced peak memory is distilled from the teacher network. For image classification, our method outperforms existing response-based and feature-based distillation methods in terms of accuracy-memory trade-off. For diffusion-based image generation, our method significantly reduces the peak memory of the denoising network with slight degradation in the fidelity and diversity of the generated images. While we focus on distillation of CNNs in this work, we leave as future work peak memory reduction of vision transformer via distillation for image classification (Wu et al. (2022)) and image generation (Peebles & Xie (2023)).

**Acknowledgement**   This work is supported by Department of Defense under funding award W911NF-241-0295, National Science Foundation Career award #2341039, and Google Cloud research credits program.

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
