# OpenReview forum: "ReDistill: Residual Encoded Distillation for Peak Memory Reduction of CNNs"
_TMLR — Accepted by TMLR_

### Review · Reviewer_zkR4 · 2025-02-13

**Summary Of Contributions:**

The paper proposes a knowledge distillation method Residual Encoded Distillation (ReDistill) to reduce the peak memory of visual models during inference, specifically for resource-constrained edge devices. This work uses aggressive downsampling in the student network’s first pooling layer and introducing Residual Encoded Distillation (RED) blocks that align feature maps between the teacher and student networks. The proposed method is evaluated on tasks such as image classification and diffusion-based image generation, achieving significant reductions in peak memory usage with minimal performance degradation.

**Audience:**

Yes

**Broader Impact Concerns:**

The reviewer does not have any broader impact concerns.

**Claims And Evidence:**

Yes

**Requested Changes:**

Please see the Weaknesses and Minor suggestions above. In particular, it is encouraged to include some more explanation about the RED computational overhead, and results on newer datasets.

**Strengths And Weaknesses:**

## Strengths

1. The study of peak memory reduction is meaningful in practice. The proposed approach, based on aggressive downsampling, is novel and intuitive.

2. The proposed approach achieves significant peak memory reduction.

3. Both image classification and generation tasks are considered and evaluated.

## Weaknesses

1. Overhead of RED modules: As shown in Table 6, students with RED may have up to 2x model size, 2x power consumption, and 3x latency, compared with the models without RED. Although the main focus of this work is the peak memory, overheads in other aspects should also be considered.

2. Low-resolution image datasets: Many of the datasets tested in this work are a bit old and contain low-resolution images, such as STL, CIFAR-10, and Celeb-A. They may no longer be representative of modern mobile applications. It would be better to train and evaluate models on more modern datasets such as iNaturalist (for classification) and CelebA-HQ (for generation). Also, high-resolution images may more significantly show the advantage of RED in reducing the peak memory.

3. Unintuitive module design: Although the reviewer can understand the RED architecture (Section 3.3) and find its effectiveness in the ablation study (Table 7), the model design still seems to be arbitrary. Why is the residual activation function ReLU6 (not regular ReLU)? Why does one convolution layer have a 3x3 kernel while the other has 1x1? Why is BatchNorm employed in RED rather than other normalization layers (e.g., LayerNorm seems more conventional when there is a vision transformer). A few sentences that explain the authors’ intuition may be helpful in the paper.

4. Effectiveness in ViT-based image classification: Table 4 only shows the accuracy comparison. Without details about computation resource consumptions, this section seems incomplete. The reviewer agrees that it might be good to investigate peak memory reduction for transformers in the future.

## Minor suggestions
1. In Section 3.2, the dimension of the teacher feature map at the $k$-th pooling layer should be $\frac{H}{\rho^k}\times\frac{W}{\rho^k}\times C_k$. The student’s initial stride and layer indices may also need to be revised.

2. Citing format: When the referred work appears as a part of the sentence, it might be better to drop the parenthese. For example in Section 3.1:
    - “Reference (Hinton et al. (2015)) defines…” - > “Hinton et al. (2015)) define”
    “specifically a U-Net network as in (Ho et al. (2020))” -> “specifically a U-Net network as in Ho et al. (2020)”

---

> ### Author Response · Authors · 2025-03-09
> **Response to Reviewer zkR4**
>
> We appreciate reviewer zkR4’s recognition of the significance of our work, the outperformance of our method
> in reducing peak memory consumption, and our comprehensive experiments. Below, we address the raised
> weaknesses.
>
> **1. Q: Overhead of RED modules.**
>
> A: One reason for significant overhead for some networks observed by the reviewer is that it requires multiple RED modules with less aggressive pooling, e.g., ×2 pooling in our framework. For aggressive pooling with much reduced peak memory, e.g, ×8 pooling, less RED blocks are introduced, leading to negligible computational overhead. Table 5 in the updated submission shows an example of distillation of MbNetV2 with ×8 pooling, for which RED modules introduced little overhead.
>
> We would like to emphasize that the focus here is on reducing peak memory with little performance degradation. Our method can be directly combined with orthogonal methods to reduce model size reduction, power consumption reduction, etc. For example, as reported in Table 10 in the updated submission, the teacher model ResNext18 achieves 85.12% accuracy, 5.00 MB peak memory, and 21.47 MB model size. The student model ResNext18×4 achieves 79.07%, 0.75 MB peak memory, and 21.47 MB model size. When applying our distillation method with mixed precision post-training-quantization method, the optimized student model achieves 84.78% accuracy, 0.44 MB peak memory, and 5.6 MB model size.
>
> **2. Q: Low-resolution image datasets.**
>
> A: We thank the reviewer’s suggestion of experiments on higher resolution datasets, for which memory constraint is a pressing issue. However, we note that the recent iNaturalist dataset is not widely used for knowledge distillation. We are not aware of any KD baselines that reported results on this dataset (Hinton et al. (2015); Adriana et al. (2015); Park et al. (2019); Jin et al. (2023); Sun et al. (2024); Hao et al. (2023)). For image generation, our teacher model and student model are based on the original DDPM (Ho et al. (2020)) implementation, which is based on pixel space instead of latent space like LDM (Rombach et al. (2022)). Training DDPM in pixel space for the CelebA-HQ dataset at such a high resolution of 1024 × 1024 is prohibitively time-consuming. Due to time constraints in the rebuttal period, we are not able to finish experiments on iNaturalist or CelebA-HQ. We will include such results in future versions of our draft once training is finished.
>
> **3. Q: Unintuitive module design.**
>
> A: We thank the reviewer for suggesting other design choices. We use ReLU6 to bound activations to prevent exploding gradients. Besides, ReLU6 is widely used in efficient neural network design (Sandler et al. (2018); Howard et al. (2019); Lin et al. (2020)) since it is particularly useful for fixed-point or low-precision inference in quantization. For kernel size in RED module, we choose 1 × 1 kernel for the logit module (LM), because LM is a gating mechanism to select critical components of the activations. We choose BatchNorm rather than LayerNorm because the proposed framework is primarily used for the distillation of CNNs, for which BatchNorm is more common.
>
> **4. Q: Effectiveness in ViT-based image classification.**
>
> A: As also pointed out by reviewer fHB7, the discussion on peak memory reduction of transformers lacks clarity, and rigorous evaluation is needed. We realized that through investigation of memory consumption for transformers for image classification and more recent DiT-based diffusion model requires substantial effort and remains an open research problem. Hence, we decided to follow reviewer fHB7’s suggestion to remove experiments on transformers and focus on CNN in this work. We have discussed future work of ViT distillation in Section 5 Conclusions and Future Work.
>
> We have fixed the typo in the notation and the reference formatting.

---

> > ### Author Response · Authors · 2025-03-09
> > **References for Response**
> >
> > Geoffrey Hinton, Oriol Vinyals, and Jeff Dean. Distilling the knowledge in a neural network. arXiv preprint arXiv:1503.02531, 2015.
> >
> > Romero Adriana, Ballas Nicolas, K Samira Ebrahimi, Chassang Antoine, Gatta Carlo, and Bengio Yoshua. Fitnets: Hints for thin deep nets. In Proceedings of the International Conference on Learning Representations, 2(3):1, 2015.
> >
> > Wonpyo Park, Dongju Kim, Yan Lu, and Minsu Cho. Relational knowledge distillation. In Proceedings of the IEEE/CVF conference on computer vision and pattern recognition, pp. 3967–3976, 2019.
> >
> > Ying Jin, Jiaqi Wang, and Dahua Lin. Multi-level logit distillation. In Proceedings of the IEEE/CVF Conference on Computer Vision and Pattern Recognition, pp. 24276–24285, 2023.
> >
> > Shangquan Sun, Wenqi Ren, Jingzhi Li, Rui Wang, and Xiaochun Cao. Logit standardization in knowledge distillation. In Proceedings of the IEEE/CVF Conference on Computer Vision and Pattern Recognition, pp. 15731–15740, 2024.
> >
> > Zhiwei Hao, Jianyuan Guo, Kai Han, Yehui Tang, Han Hu, Yunhe Wang, and Chang Xu. One-for-all: Bridge the gap between heterogeneous architectures in knowledge distillation. In Advances in Neural Information Processing Systems, 2023.
> >
> > Jonathan Ho, Ajay Jain, and Pieter Abbeel. Denoising diffusion probabilistic models. Advances in neural information processing systems, 33:6840–6851, 2020.
> >
> > Robin Rombach, Andreas Blattmann, Dominik Lorenz, Patrick Esser, and Björn Ommer. High-resolution image synthesis with latent diffusion models, 2022. URL https://arxiv.org/abs/2112.10752.
> >
> > Mark Sandler, Andrew Howard, Menglong Zhu, Andrey Zhmoginov, and Liang-Chieh Chen. Mobilenetv2: Inverted residuals and linear bottlenecks. In Proceedings of the IEEE conference on computer vision and pattern recognition, pp. 4510–4520, 2018.
> >
> > Andrew Howard, Mark Sandler, Grace Chu, Liang-Chieh Chen, Bo Chen, Mingxing Tan, Weijun Wang, Yukun Zhu, Ruoming Pang, Vijay Vasudevan, et al. Searching for mobilenetv3. In Proceedings of the IEEE/CVF international conference on computer vision, pp. 1314–1324, 2019.
> >
> > Ji Lin, Wei-Ming Chen, Yujun Lin, Chuang Gan, Song Han, et al. Mcunet: Tiny deep learning on iot devices. Advances in Neural Information Processing Systems, 33:11711–11722, 2020.

---

### Review · Reviewer_NDk8 · 2025-02-23

**Summary Of Contributions:**

This paper proposed residual encoded distillation (ReDistill) for peak memory reduction and efficient deployment of large-scale models on resource-constrained devices. While most previous knowledge distillation work focuses on model size compression or improving the performance of a given model with a larger one, ReDistill is the first targeting peak memory reduction. ReDistill derives the lightweight student model from the teacher network using aggressive pooling and a residual encoded distillation (RED) block to align features between teacher and student. The proposed ReDistill is extensively verified on various architectures (CNNs, ViTs, Diffusion Models) and tasks (classification, generation), demonstrating effectiveness and efficiency.

**Audience:**

Yes

**Claims And Evidence:**

Yes

**Requested Changes:**

- Some typos and grammar errors could be fixed:
  - Sec.1 ...a less accuracy drop $\rightarrow$ ...a lower accuracy drop
  - Sec.4.2 all models 250K iterations $\rightarrow$ all models with 250K iterations
  - Tab.6 Actural $\rightarrow$ Actual, 2.8 $\rightarrow$ 2.80
  - ...

**Strengths And Weaknesses:**

Strengths:
- Reducing peak memory is a good research topic considering the memory-bound scenario of current large-scale transformer-based model deployment.
- The presentation quality is satisfying. The target and architecture of the proposed ReDistill framework are introduced clearly.
- Comprehensive experiments are conducted on various models and tasks. Especially the image generation with diffusion model is not included in most previous work but is highlighted in this paper.
- Edge device deployment results align with the theoretical peak memory analysis, improving the credibility of results and robustness of this work.

Weakness:
- KD baselines in the experiments are somewhat outdated, please consider including more recent KD baselines in the experiments such as [1] and [2].

[1] One-for-All: Bridge the Gap Between Heterogeneous Architectures in Knowledge Distillation, NeurIPS 2023

[2] Logit Standardization in Knowledge Distillation, CVPR 2024

---

> ### Author Response · Authors · 2025-03-09
> **Response to Reviewer NDk8**
>
> We appreciate that reviewer NDk8 find our work a good research topic, our presentation clear, and our
> experiments comprehensive. We address the critical feedback in the following.
>
> **1. Q: consider including more recent KD baselines in the experiments**
>
> A: We thank reviewer NDk8’s suggestion about recent KD baselines, including LSKD (Sun et al. (2024)) and OFAKD (Hao et al. (2023)). We report these two baselines’ metrics on ImageNet dataset in Table 1 in the updated submission. For ResNet18×4 distilled from ResNet18, LSKD and OFAKD achieve 63.81% and 64.83%, respectively, while our method achieves 65.23%. For ResNet50×4 distilled from ResNet50, LSKD achieves 72.28%, while our method achieves 73.23%. For MobileNetV2×2 distilled from ResNet152, LSKD and OFAKD achieve 66.21% and 68.49%, respectively, while our method achieves 68.89%. The experiments demonstrate that our method outperforms these two baselines under the aggressive pooling settings distillation.
>
> Thanks for identifying typos, which we have fixed in the updated submission.
>
> **References:**
>
> Shangquan Sun, Wenqi Ren, Jingzhi Li, Rui Wang, and Xiaochun Cao. Logit standardization in knowledge distillation. In Proceedings of the IEEE/CVF Conference on Computer Vision and Pattern Recognition, pp.15731–15740, 2024.
>
> Zhiwei Hao, Jianyuan Guo, Kai Han, Yehui Tang, Han Hu, Yunhe Wang, and Chang Xu. One-for-all: Bridge the gap between heterogeneous architectures in knowledge distillation. In Advances in Neural Information Processing Systems, 2023.

---

### Review · Reviewer_fHB7 · 2025-02-23

**Summary Of Contributions:**

The authors introduce a new method called "residual encoded distillation" (ReDistill) that allows distilling a CNN teacher network into a CNN student network with 4-5x reduced theoretical peak memory requirements, with the goal to make the models more suitable for deployment on memory-constrained devices.

Their method is based on a more aggressive initial pooling, and the addition of a "residual encoded distillation (RED)" block for feature distillation.

The authors claim improvements for image classification on ImageNet and STL10, as well as for image generation for CIFAR10 and Celeb-A (using DDPM).

**Audience:**

Yes

**Broader Impact Concerns:**

None.

**Claims And Evidence:**

No

**Requested Changes:**

Critical changes:

1. Better ablations: I would be interested to see how different RE kernel sizes perform, and similarly for alpha values different from 0, 1, and 50 (is 50 optimal? does the same optimal value work equally well for every architecture and pooling reduction factor?).

2. Explain how exactly the image classification models were configured. I could find no details on the "aggressive pooling setting", other than the "first pooling layer stride ×2 ∼ ×8 times and adjust the last several pooling layer strides to ensure the final output of the student model with the same information density as the teacher model" (Section 4.2). How exactly were the networks configured? I thought (after reading Section 3.2 and Figure 2) that the pooling layers were all the same, other than the initial pooling layer that incorporates multiple pooling layers from the teacher architecture. Also, it's not clear what the layers correspond to in Figure 6 (e.g. there are only 9 layers shown for ResNet18, and 17 layers for ResNet50; also, every layer has lower memory footprint, even though the previous description only talks about memory footprint reduction in the initial layers). Why the decision to look at "x4" variants for the classification of the experiments ("x2" is only mentioned once in Tables 1-3) and "x2" for the generation experiments? In this context it would also be helpful to explain how the peak memory reduction is computed?

3. Fix the discrepancy of the CIFAR10 FID in Table 5 (7.23 vs 3.17 mentioned in [(Ho, 2020)]).

4. Compare ReDistill with other distillation methods in image generation (Section 4.5), or explain why this is not possible.

5. I find the values in Table 3 surprising: Why is ReDistill compared to a set of distillation methods different from Tables 1 and 2? How come that many of the reported values are *worse* than the student architecture without distillation? Please explain this better in the main text, or remove the table.

Medium importance changes:

1. I think the presented method in its current form only makes sense for CNNs, so I think it would help the reader and add clarity to mention "convolution" already in the title.

2. I don't find that the section about ViTs adds anything to the paper. If the authors want to keep the section, then I would like to see my points above (under weaknesses) addressed.

3. I find it confusing that the RED block (in Figure 2) has $f^T$ as an *input*, since it is only used to compute the loss. I would recommend having the RED block with simply $f^S$ and $f^D$ as input and output (ideally using some notation that makes it clear that they have the same dimensions), and then have an arrow going from $f^D$ to box $L_{RED}$ (with another arrow coming from $f^T$).

4. I find Figure 2 hard to parse: Why does it list stages 1,k,k+1,n for the teacher and stages 1,2,i,i+1 for the student? From the text I believe to understand that only stages 1..k-1 are skipped and from stage k on, every stage is found in both teacher and student (so no need to list stage n). Also, the text says that teacher and student stages have the same spatial dimensionality in the output, but might be different wrt number of channels – but the figure is exactly the other way around (i.e. illustration of teacher and student feature maps have the same number of channels, but different size).

5. Explain why applying QAT (Table 11) only saves 10% in model size and compare QAT with the presented method in terms of peak memory reduction – theoretical, and ideally also practical. I think that comparing ReDistill with other methods mentioned in related work "memory-constrained deep learning" would make the submission more practically relevant.

[(Ho, 2020)]: https://arxiv.org/abs/2006.11239


Minor comments:

1. In the introduction, the authors motivate their approach with a STM32H5 MCU (640 kb of RAM), but then in the practical experiments, they run the networks on a NVIDIA Jetson TX2 (8GB of RAM) – it seems that the presented method is not at all applicable to the kind of edge device mentioned in the introduction.
2. "However, we still keep the same spatial dimensions of the input activations to the final fully connected layer." (section 3.2) – what exactly is kept the same here? Is it the spatial dimensions of the inputs (i.e. the input image), or the spatial dimensions of the teacher (which is the input to the student during distillation). Please clarify.
3. "Homogeneously" (section 3.2) – consider writing "similarly", or "correspondingly" instead
4. Equations (5,6,7) – why not reuse symbols from Figure 3 and text? i.e. $f^R=r(·)$ and $\hat{f}^S=l(·)$.
5. Table 6 mentions MbNetV2x8 but table Table 2 only has results for MbNetV2x4 – maybe a typo?

**Strengths And Weaknesses:**

Strengths:

1. The authors introduce a new component, the RED block, that significantly improves the performance of a student with lower memory requirements (4x in theory, 2x on a NVIDIA Jetson).

2. The presented method shows improvements across both image classification and generation.

3. In the case of classification (but not generation), the authors compare their method with many other established distillation methods.

Weaknesses:

1. Section 4.4 "ViT-based Image Classification": since the RED module "is designed to ensure that the output of a pooling layer closely resembles the distribution of the input while preserving essential features at a reduced spatial dimension", it is not clear why it is even applied on the ViT architecture. In the case of the ViT, there is no such reduced spatial dimension, from what I can tell, and also is there no reduction in theoretical peak memory requirement. It also seems contra-intuitive to train a base-sized transformer on STL10 (only 5k training examples!), but to use the tiny-sized transformer for ImageNet (with 1.2M training examples). Note that [(Touvron, 2020)] reports 74.5 for ViT-Ti/16 distilled, and 72.2 for ViT-Ti/16 from scratch in table 5 (vs. 72.5 reported here for both distillation with and without RED).

2. I do appreciate the comparison with many other distillation methods, but I would really like to see the results anchored in some previously published baselines. From what I can tell, tables 1-3 don't report numbers from other papers, and table 5 reports a FID=7.23 on CIFAR10 for the teacher network, but when looking at [(Ho, 2020)] Table 1 I find FID=3.17 (note that the text in Section 4.5 says "results slightly different from those in the original paper", but this difference is unacceptably large, assuming that I'm indeed comparing to the correct value).

3. The authors mention related work for "memory-constrained deep learning", such as quantization, but then the authors only compare existing distillation methods in their setup (with more aggressive pooling). I could only find a short mention of QAT in Section 4.7, and Table 11 mentions model size (which is reduced by only 10%, so I wonder what kind of quantization was used), but not peak memory requirement.

4. Even though the paper is well written overall, I found some of parts hard to parse, and some important details missing (see requested changes below).

[(Touvron, 2020)]: https://arxiv.org/pdf/2012.12877
[(Ho, 2020)]: https://arxiv.org/abs/2006.11239

---

> ### Author Response · Authors · 2025-03-09
> **Response to Reviewer fHB7 part 1/3**
>
> We thank the reviewer fHB7’s thorough and critical feedback. Below, we address concerns raised by the reviewer. We have made almost all the required changes in the updated version.
>
> **1. Q: Better ablation on kernel size and alpha values**
>
> A: We have conducted ablation studies on kernel size and alpha values, as reported in Table 6 and Table 7 in the updated version. We initially used α = 50 in our experiments. We further tried more α values and obtained slightly better results with α = 100. We increase the kernel size of the residual module of the proposed RED block from 1 to 5 and find that it achieves the best performance with kernel size 3.
>
> **2. Q: Explain how exactly the image classification models were configured, why the decision to look at "x4" variants for the classification of the experiments and "x2" for the generation experiments, and how the peak memory reduction is computed.**
>
> A: We show the network config details in Figure 5 of the updated submission for our ‘T: ResNet18’ and ‘S: ResNet18×4’ pair used in the third column of Table 1. As shown in Figure 5, we aggressively increase the stride of the first downsampling layer ‘Conv 1’ from 2 to 8 following the aggressive pooling setting, while setting the stride to 1 of maxpool layer and the last downsampling layer, i.e., the first conv layer in ‘Stage 4’, for keeping the same activation size for average pool and fc layer. Then the student network possesses only three downsampling layers less than five downsapling layers of the teacher network. We integrate the proposed RED block after each downsampling layer of the student in our ReDistill framework to improve the student’s performance while maintaining low peak memory.
>
> The reason that we report "x4" variants for the classification of the experiments and "x2" for the generation experiments is that the input resolution is 128 × 128 or 224 × 224 for image classification, while 64 × 64 for image generation. Such an aggressive pooling setting could provide relatively unified intermediate activations’ resolution for classification and generation.
>
> In Figure 7, we report the local maximum memory consumption of the network layers inner each grid cell in the 2nd and 6th columns of Figure 5 as the ResNet18 baseline and our method’s peak memory footprint. The peak memory reduction is calculated by subtracting the maximum memory in the memory footprint, as shown in Figure 7 of the baseline and our method.
>
> **3. Q: Fix the discrepancy of the CIFAR10 FID**
>
> A: In the updated submission, we retrain DDPM models on CIFAR10 with more iterations to get a similar FID as reported in Ho et al. (2020). As shown in Table 4 in the updated submission, the teacher U-Net "T: U-Net w/ DDPM" achieves 3.75 FID, similar to the FID 3.17 reported in Ho et al. (2020). We also retrain the student networks with or without our proposed distillation method.
>
> **4. Q: Compare ReDistill with other distillation methods in image generation**
>
> A: Current distillation methods (Salimans & Ho (2022); Yin et al. (2024a;b)) for DDPM mainly focus on reducing the time steps instead of optimizing students’ overheads, which is orthogonal to conventional distillation methods in image classification and also our method. These methods don’t reduce the peak memory of DDPM. Hence, it is not meaningful to compare our method with these distillation methods for peak memory reduction.
>
> Following the reviewer’s suggestion, we implemented a simple distillation method, which we termed MSE-Distill-DDPM, by directly matching the final output activations of the teacher model and student model using MSE loss. As shown in Table 4 in the updated version, on CIFAR10 dataset, the teacher U-Net achieves 3.75 FID and 9.70 IS score, while the student U-Net×2 without distillation achieves 14.28 FID and 9.12 IS score. The student with MSE-Distill achieves 12.04 FID and 9.62 IS score, while our method achieves the best 10.85 FID and 9.63 IS score. This experiment demonstrates that our method outperforms MSE-Distill-DDPM with reduced peak memory and better image quality.

---

> > ### Author Response · Authors · 2025-03-09
> > **Response to Reviewer fHB7 part 2/3**
> >
> > **5. Q: Why is ReDistill compared to a set of distillation methods different from Tables 1 and 2?**
> >
> > A: We are not able to compare to some KD methods on STL10 dataset because these KD methods requires a customized dataloader, like CRD (Tian et al. (2020)), MLLD (Jin et al. (2023)), which require a contrastive version of the input data, but these methods don’t provide corresponding dataloader for STL10 dataset in their original code implementations. Besides, we don’t have enough computation resources to conduct experiments on ImageNet to compare with all KD methods reported in Table 2 and 3. We have to select some representative KD methods on ImageNet, like KD (Hinton et al. (2015)), FitNet (Adriana et al. (2015)), and RKD (Park et al. (2019)), which are reported in all three Tables 1 ∼ 3.
> >
> > **6. Q: Tables 1 ∼ 3 don’t report numbers from other papers.**
> >
> > A: Our results of the teacher networks and student networks match with what is reported in previously published papers. For example, our ResNet50 teacher network achieves a top-1 accuracy of 76.13% on ImageNet shown in Table 1, which is comparable to the 76.16% reported in Table 4 of MLLD (Jin et al. (2023)). Our ResNet18 teacher network, i.e., the standard ResNet18 network, achieves the top-1 accuracy of 69.75% on ImageNet dataset as reported in Table 1, while the standard ResNet18, as a student model w/o distillation reported in MLLD, achieves the same 69.75% top-1 accuracy in Table 4 of MLLD (Jin et al. (2023)).
> >
> > However, it doesn’t make sense to report numbers directly from or anchored in other papers for distilled network, because we actually conduct the distillation differently from their original settings. In these conventional KD baseline’ settings, the student network doesn’t have to possess a lower peak memory than the teacher network. For a representative instance, a student network ResNet18 is distilled from a teacher network ResNet34 as shown in Table 4 of MLLD (Jin et al. (2023)). Under
> > this setting, the peak memory of the student and teacher are the same 3.83 MB, even though the model size of the student is reduced. (ResNet18 and ResNet34 have the same basic block with the same width but have a different number of layers. Only the specific layer with maximum memory consumption affects the peak memory consumption when having the same inputs, that is, the difference of the number of layers between these two networks won’t affect the peak memory.)
> >
> > In our aggressive pooling settings, our student network ResNet18×4 is derived from the standard network ResNet18 by increasing the first convolution layer’s stride 4 times, significantly reducing the peak memory of ResNet18 from 3.83 MB to 0.77 MB. Even if we keep the same student-teacher network pair, i.e., ResNet18-ResNet34, we actually build a student network ResNet18×4 with lower peak memory, different from the standard network ResNet18 reported in previously published
> > baselines.
> >
> > **7. Q: How come that many of the reported values are worse than the student architecture without distillation?**
> >
> > A: The reason that some reported values are worse than the student architecture without distillation is some what mentioned in our response above. We actually conduct the distillation differently from these KD baselines’ original settings. Under such an aggressive pooling setting, traditional distillation methods are not guaranteed to be positively effective, especially for feature-based distillation methods. Generally, conventional feature-based methods match the activations of the teacher model and student model stage by stage. In each stage, these activations possess the same resolution. However, in our aggressive pooling setting, the resolution of student activations and teacher activations are mismatched in each stage, and thus conventional KD methods might fail to transfer information stage by stage. This also illustrates the importance and necessity of the proposed ReDistill framework.

---

> > > ### Author Response · Authors · 2025-03-09
> > > **Response to Reviewer fHB7 part 3/3**
> > >
> > > **8. Q: the presented method in its current form only makes sense for CNNs. I don’t find that the section about ViTs adds anything to the paper**
> > >
> > > A: We sincerely consider the reviewer’s suggestion, and due to the limited time for conducting supplementary experiments on Vision Transformer, we agree that the current Section 4.4 "ViT-based Image Classification" is less related to the main topic of this paper. The proposed ReDistill framework mainly works for convolutional neural networks and can not reduce the peak memory of transformer-based networks. Thus, we decided to remove section 4.4 and leave a thorough investigation of peak memory reduction for transformers as future work.
> > >
> > > We have changed the title to include "CNN".
> > >
> > > **9. Q: Explain why applying QAT (Table 11) only saves 10% in model size and compare QAT with the presented method in terms of peak memory reduction**
> > >
> > > A: Regarding the concern about Table 11 of our original submission, the comparison to other quantization methods, we deployed quantization-aware-training (QAT) of TorchAO (torchao maintainers & contributors (2024)), which is the PyTorch native architecture optimization toolkit. We found it is because TorchAO only supports dynamic quantization for linear layers while excluding convolutional layers, which means only the fully connected layer is quantized in the model. In the updated submission, we apply Intel Neural Compressor (INC) Cheng et al. (2023), a neural network compression toolkit, widely used in related research, like Cheng et al. (2024); Shen et al. (2022), to conduct post-training-quantization (PTQ) for the network. Besides manually setting the quantization policy, INC can also automatically search for the optimal mixed precision quantization policy for the network.
> > >
> > > As shown in Table 10 of the updated submission, the mixed precision quantized ResNext18 achieves a model size of only 5.015 MB, representing a 76.6% reduction compared to the 21.47 MB size of the fp32 model. For the mixed precision PTQ, INC found an optimal quantization policy to quantize almost all the layers’ weights and activations to int8 but keep the first convolution layer and ReLU function as fp32, that is, the peak memory for this mixed precision quantized ResNext18 is caused by the fp32 activations of the first convolution layer, 4.19 MB, as reported in Table 10. We also compare with a 6 bits QAT method, INQ (Aojun Zhou (2017)), which achieves the lowest model size of 4.026 MB, as shown in Table 10. However, as a weight-only quantization method, INQ can not reduce the peak memory caused by the high-resolution activations.
> > >
> > > In summary, the ResNext18 quantized by INC to int8 achieves 20.12% accuracy and 1.25 peak memory. The ResNext18 quantized by INC to mixed precision achieves 84.18% accuracy and 4.19 MB peak memory. The ResNext18 quantized by INQ to 6 bits achieves 84.07% accuracy and 5.00 MB peak memory. Our distillation method achieves the optimal trade-off between accuracy and peak memory, with 84.80% accuracy and 0.75 MB peak memory. Due to the extra RED blocks, our method possesses the higher model size of 25.557 MB. However, our method is orthogonal to these quantization methods and thus can be applied with the quantization method to further optimize the student model’s overhead. As shown in the last row of Table 10, the student model ResNext18×4 distilled by our method and quantized by INC to mixed precision achieves 84.78% accuracy, 0.44 MB peak memory, and 5.621 MB model size.
> > >
> > > **10. Q: Figure 2 hard to parse.**
> > >
> > > A: We have optimized Figure 2 according to the reviewer’s suggestion in our updated submission. We also show the network config and aggressive pooling setting in detail in Figure 5 of the updated submission, which is explained in our response 2. We hope it can help to understand our ReDistill framework.
> > >
> > > Thanks for the reviewer’s other minor comments. We have made corresponding changes in the updated version.

---

> > > > ### Author Response · Authors · 2025-03-09
> > > > **References for Response**
> > > >
> > > > Jonathan Ho, Ajay Jain, and Pieter Abbeel. Denoising diffusion probabilistic models. Advances in neural information processing systems, 33:6840–6851, 2020.
> > > >
> > > > Tim Salimans and Jonathan Ho. Progressive distillation for fast sampling of diffusion models. arXiv preprint arXiv:2202.00512, 2022.
> > > >
> > > > Tianwei Yin, Michaël Gharbi, Taesung Park, Richard Zhang, Eli Shechtman, Fredo Durand, and William T Freeman. Improved distribution matching distillation for fast image synthesis. arXiv 2405.14867, 2024a.
> > > >
> > > > Tianwei Yin, Michaël Gharbi, Richard Zhang, Eli Shechtman, Frédo Durand, William T Freeman, and Taesung Park. One-step diffusion with distribution matching distillation. In CVPR, 2024b.
> > > >
> > > > Yonglong Tian, Dilip Krishnan, and Phillip Isola. Contrastive representation distillation. In International Conference on Learning Representations, 2020.
> > > >
> > > > Ying Jin, Jiaqi Wang, and Dahua Lin. Multi-level logit distillation. In Proceedings of the IEEE/CVF Conference on Computer Vision and Pattern Recognition, pp. 24276–24285, 2023.
> > > >
> > > > Geoffrey Hinton, Oriol Vinyals, and Jeff Dean. Distilling the knowledge in a neural network. arXiv preprint arXiv:1503.02531, 2015.
> > > >
> > > > Romero Adriana, Ballas Nicolas, K Samira Ebrahimi, Chassang Antoine, Gatta Carlo, and Bengio Yoshua. Fitnets: Hints for thin deep nets. In Proceedings of the International Conference on Learning Representations, 2(3):1, 2015.
> > > >
> > > > Wonpyo Park, Dongju Kim, Yan Lu, and Minsu Cho. Relational knowledge distillation. In Proceedings of the IEEE/CVF conference on computer vision and pattern recognition, pp. 3967–3976, 2019.
> > > >
> > > > Ying Jin, Jiaqi Wang, and Dahua Lin. Multi-level logit distillation. In Proceedings of the IEEE/CVF Conference on Computer Vision and Pattern Recognition, pp. 24276–24285, 2023.
> > > >
> > > > torchao maintainers and contributors. torchao: Pytorch native quantization and sparsity for training and inference, October 2024. URL https://github.com/pytorch/torchao.
> > > >
> > > > Wenhua Cheng, Yiyang Cai, Kaokao Lv, and Haihao Shen. Teq: Trainable equivalent transformation for quantization of llms, 2023. URL https://arxiv.org/abs/2310.10944.
> > > >
> > > > Wenhua Cheng, Weiwei Zhang, Haihao Shen, Yiyang Cai, Xin He, Kaokao Lv, and Yi Liu. Optimize weight rounding via signed gradient descent for the quantization of llms, 2024. URL https://arxiv.org/abs/2309.05516.
> > > >
> > > > Haihao Shen, Ofir Zafrir, Bo Dong, Hengyu Meng, Xinyu Ye, Zhe Wang, Yi Ding, Hanwen Chang, Guy Boudoukh, and Moshe Wasserblat. Fast distilbert on cpus, 2022. URL https://arxiv.org/abs/2211.07715.
> > > >
> > > > Aojun Zhou, Anbang Yao, Yiwen Guo, Lin Xu, Yurong Chen. Incremental network quantization: Towards
> > > > lossless cnns with low-precision weights. In International Conference on Learning Representations,ICLR2017,
> > > > 2017.

---

> > > > > ### Comment · Reviewer_fHB7 · 2025-03-20
> > > > >
> > > > > I'd like to thank the authors for the many updates in the submission.
> > > > >
> > > > > The revisions have indeed addressed all my major remarks, and the paper is now much clearer and draws a stronger picture of the proposed method. I particularly appreciate the added Figure 5 (which makes Figure 2 and Figure 7 a lot easier to understand) and the much improved section on quantization. I also think the updated revision is more convincing without the section about ViTs and calling out CNN in the title (nit: I think "CNNs" would read more naturally).
> > > > >
> > > > > Some minor comments on the updated revision:
> > > > >
> > > > > 1. The "S + MSE-Distill" row is missing for Celeb-A in Table 4.
> > > > >
> > > > > 1. The technical reason why some distillation methods could not be applied to STL10 should be mentioned in the paper (I think other readers will have the same question as I did when comparing Tables 1 and 2).
> > > > >
> > > > > 1. Similarly, the fact that "Under such an aggressive pooling setting, traditional distillation methods are not guaranteed to be positively effective" is not obvious and it might be useful to mention that in the text as well.

---

### Review · Reviewer_TGXm · 2025-02-26

**Summary Of Contributions:**

This paper proposes a novel model distillation method for computer vision tasks that emphasizes minimizing peak memory consumption. The approach employs aggressive pooling combined with newly proposed RED blocks and is optimized using a dedicated loss function. The experimental results suggest that RED-based distilled models achieve competitive accuracy while maintaining bounded peak memory usage.

**Audience:**

Yes

**Claims And Evidence:**

Yes

**Requested Changes:**

It is recommended that the authors provide more detailed memory consumption metrics—both theoretical estimates and empirical measurements—for all compared techniques (e.g., NST, DeiT). This clarification could be incorporated into Table 6 or presented in an additional section to better contextualize the comparative performance.

**Strengths And Weaknesses:**

Strengths

The RED student models demonstrate top performance across various computer vision tasks.

The manuscript is well written and easy to follow.


Weaknesses

Table 6 indicates that in some cases, the actual peak memory of RED-based models exceeds that of other student techniques.

---

> ### Author Response · Authors · 2025-03-09
> **Response to Reviewer TGXm**
>
> We appreciate reviewer TGXm's comments and the recognition of our method’s top performance across various computer vision tasks.
>
>
> **1\. Q: the actual peak memory of RED-based models exceeds that of other student techniques.**
>
> A: We thank the reviewer for pointing this out. We found the extra peak memory is due to our suboptimal implementation. To integrate RED blocks into any network, we utilized Pytorch Hooks to maintain a lookup table to trace intermediate activations of student networks, which results in additional memory consumption. We found this additional memory accounts for about 10% of the student’s peak memory and can be eliminated by hardcoding RED blocks into the student networks rather than using lookup tables during inference. We further hardcode the RED blocks into each student network and re-measure the actual peak memory. Our updated Table 5 shows that we achieve the same actual peak memory consumption with the student network.
>
>
>
>
> **2\. Q: provide more detailed memory consumption metrics...This clarification could be incorporated into Table 6 or presented in an additional section.**
>
> A: Firstly, let us clarify how theoretical peak memory is computed. We follow previous work, MCUNet (Lin et al. (2020; 2021; 2022)), for calculating peak memory. Generally, it traces the memory consumption for each layer. For the standard convolutional layer, the maxpool layer, or the average pool layer, the memory consumption is the summation of the input activation memory size and output activation memory size. For a group convolutional layer, the memory consumption is the summation of the input activation memory size, the output activation memory size, and a buffer with size equal to one channel convolutional kernel in the group. For the residual connection in the network, the residual activations memory size will be added into memory tracing until the residual item is added to the output activations. We have added the description of peak memory computation in Section 4.2 on implementation details.
>
> Secondly, we discuss the memory consumption of other distillation methods. Response-based KD methods generally keep the same peak memory with the student model, since they don’t alter the student architecture. For feature-based KD methods, some might increase the student’s peak memory due to extra trainable modules. However, none of these KD methods can lead to peak memory lower than the student, which is a lower bound. Our method reaches such a lower bound and achieves the highest accuracy compared to existing KD methods as shown in our comprehensive experiments. We have added the discussion of memory consumption for distillation methods in Section 3.1.
>
> For the requested change about memory consumption of DeiT, we follow other reviewers’ suggestion of removing experiments on ViT (Section 4.4 ViT based image classification in the original submission) and focusing only on CNN distillation. The exploration of memory reduction for ViT is left as future work, see our response to reviewer fHB7.
>
>
>
> **References:**
>
> Ji Lin, Wei-Ming Chen, Yujun Lin, Chuang Gan, Song Han, et al. Mcunet: Tiny deep learning on iot devices. Advances in Neural Information Processing Systems, 33:11711–11722, 2020.
>
> Ji Lin, Wei-Ming Chen, Han Cai, Chuang Gan, and Song Han. Mcunetv2: Memory-efficient patch-based inference for tiny deep learning. arXiv preprint arXiv:2110.15352, 2021.
>
> Ji Lin, Ligeng Zhu, Wei-Ming Chen, Wei-Chen Wang, Chuang Gan, and Song Han. On-device training under 256kb memory. Advances in Neural Information Processing Systems, 35:22941–22954, 2022.

---

### Author Response · Authors · 2025-03-09
**Updated Submission and General Response to all Reviewers**

We thank all the reviewers for their detailed, constructive and insightful feedback. It is encouraging that the reviewers find the research topic is meaningful in practice, the manuscript is well-written and easy to follow, and we show significant peak memory reduction for both image classification and image generation.

We have spent substantial efforts to further improve the clarity, e.g., on peak memory computation and on our aggressive pooling strategy by providing example architecture. We have also included additional experiments on suggested baselines, ablation study, and comparison/integration with quantization methods. We have made almost all the requested changes and have updated our manuscript. (Please note the major change in blue text in the PDF.) The only ongoing experiment that we are not able to finish during the rebuttal period is image generation on higher-resolution datasets suggested by reviewer zkR4 to further demonstrate the advantage of our method. We will update the results in future versions of our work.

We look forward to any other feedback and suggested changes.

---

### Decision · Action_Editor_rUHQ · 2025-04-05

**Recommendation:** Accept with minor revision

**Comment:**

All reviewers were positive about the work in the post-rebuttal phase, and they especially liked the proposed module and distillation method. Besides, the authors have provided extensive new experiments in response to the reviewers' suggestions. Hence, the decision is to recommend the paper for acceptance with a minor revision for incorporating the new experiments.

**Audience:**

Peak memory reduction should be relevant and useful for ML practitioners of various fields. The paper provides them a novel technique in that regard.

**Claims And Evidence:**

The paper studies an interesting problem, reducing CNNs' peak memory usage. The proposed method is distillation combined with a novel specialized loss. Reviewers all liked the approach and results, so the decision is to recommend the paper for acceptance.